# Biomolecular Correlates of Chronic Affective Dysregulation in PTSD: CDRS and Serum Markers SUMO1, MDA, CX3CL1, and UCHL1

**DOI:** 10.3390/ijms262010214

**Published:** 2025-10-21

**Authors:** Izabela Woźny-Rasała, Ewa Alicja Ogłodek

**Affiliations:** Collegium Medicum, Jan Dlugosz University in Częstochowa, Waszyngtona 4/8 Street, 42-200 Częstochowa, Poland; i.wozny-rasala@ujd.edu.pl

**Keywords:** dysthymia, fractalkine, malondialdehyde, post-traumatic stress disorder, small ubiquitin-like modifier 1, ubiquitin C-terminal hydrolase L1

## Abstract

Post-traumatic stress disorder (PTSD) is frequently comorbid with persistent depressive disorder (dysthymia), indicating shared neurobiological pathways that influence stress modulation, emotional regulation, and neurohormonal adaptation. This study examines the roles of serum biomarkers—small ubiquitin-like modifier 1 (SUMO1), malondialdehyde (MDA), fractalkine (CX3CL1), and ubiquitin C-terminal hydrolase L1 (UCHL1)—involved in oxidative stress management, neuroimmune regulation, and neuronal proteostasis. In this cross-sectional analysis, biomarker expression was assessed in 92 male trauma-exposed participants aged 19–50 years, divided into three groups: PTSD duration ≤ 5 years (n = 33, median age 34.0 years [IQR 31.0–41.0]), PTSD duration > 5 years (n = 31, median age 36.0 years [IQR 29.5–41.0]), and controls without current or past PTSD (n = 28, median age 33.5 years [IQR 24.3–41.5]). Participants were stratified into younger (19–34 years) and older (35–50 years) cohorts to account for age-related neurobiological variability. Dysthymic symptomatology was evaluated using the Cornell Dysthymia Rating Scale (CDRS), focusing on chronic subthreshold depressive features. Results indicated a significant association between PTSD and elevated dysthymic symptom burden (*p* < 0.001), with both PTSD subgroups demonstrating mild to moderate CDRS severity compared to euthymic controls. Biomarker analysis revealed phase-dependent alterations: SUMO1 levels were significantly elevated in the ≤5 years PTSD group compared to controls (*p* = 0.002), suggesting early compensatory neuroprotection, whereas UCHL1 was markedly increased in the >5 years PTSD group (*p* = 0.015), which is indicative of chronic neuronal damage and proteostatic disruption. No significant differences were observed in MDA or CX3CL1 across groups (*p* > 0.05). These findings highlight PTSD’s contribution to sustained affective dysregulation, potentially mediated by temporal shifts in oxidative stress and protein homeostasis markers. Clinically, this supports the utility of biomarker profiling for risk stratification, early intervention, and personalized therapeutic strategies, such as targeted modulation of SUMOylation or UCHL1 activity, to enhance neuroresilience and mitigate progression to severe mood disorders.

## 1. Introduction

Post-traumatic stress disorder (PTSD) is a serious mental disorder that develops in response to traumatic experiences. According to the *Diagnostic and Statistical Manual of Mental Disorders, Fifth Edition* (DSM-5), trauma involves a direct threat to life, severe injury, or being subject to violence. The diagnosis of PTSD requires the presence of intrusion symptoms (e.g., intrusive memories and nightmares), persistent avoidance of trauma-related stimuli, negative changes in mood and thinking, and increased reactivity and arousal. These symptoms must persist for at least one month and cause significant impairment in functioning across multiple areas of life [1,2,3].

Post-traumatic stress disorder can coexist with mood disorders, such as persistent depressive disorder, also known as dysthymia. Dysthymia is defined as a chronic state of low mood lasting at least two years, accompanied by symptoms such as sleep disturbances, a lack of energy, low self-esteem, difficulties in concentration, and feelings of hopelessness. At least two of these symptoms must be present most days, without remission periods longer than two months. Dysthymia is milder than major depression but has a chronic course and can significantly reduce quality of life [4].

Both PTSD and dysthymia are characterized by chronic emotional and neurobiological dysregulation, and their co-occurrence intensifies neuroinflammation.

Numerous studies [5,6,7,8] indicate that the coexistence of these disorders may lead to deepening depressive states, more difficult diagnosis, and decreased treatment effectiveness. Furthermore, shared biological features, such as dysfunction of the hypothalamic–pituitary–adrenal (HPA) axis, reduced neuroplasticity, and chronic inflammation, suggest the existence of common pathophysiological mechanisms. Both disorders show neuroanatomical changes in the hippocampus and amygdala, as well as persistent neurotransmission disturbances [9].

In recent years, growing evidence has highlighted the key role of oxidative stress and inflammatory responses in the pathogeneses of these disorders [10,11]. This article focuses on four biomarkers, Small Ubiquitin-like Modifier 1 (SUMO1), malondialdehyde (MDA), fractalkine (CX3CL1), and ubiquitin C-terminal hydrolase L1 (UCHL1), which represent complex cellular processes: proteostasis, lipid peroxidation, inflammatory signaling, and cellular stress response.

SUMO1 is one of the factors involved in the neuroinflammatory process in PTSD and mood disorders. This molecule belongs to the family of small SUMO proteins involved in the post-translational modification of nuclear and cytoplasmic proteins, affecting their stability, localization, activity, and interactions—especially those engaged in transcription and DNA repair processes [10]. In the context of PTSD and dysthymia, SUMOylation may have a protective effect on neurons exposed to prolonged stress. SUMOylation of proteins such as cyclic adenosine monophosphate (cAMP) response element-binding protein (CREB), nuclear factor kappa-light-chain-enhancer of activated B cells (NF-κB), ETS Like-1 protein (Elk-1), p53 protein, and histone deacetylase 1 (HDAC1) is crucial for cognitive and emotional processes in PTSD. Studies have shown that SUMO1 expression changes in response to chronic stress [11,12]. In depression models, increased SUMOylation has been observed and linked to compensatory mechanisms against damage due to oxidative stress and dysregulation of synaptic gene transcription [12]. Many researchers [11,12,13,14,15] suggest that SUMOylation disorders contribute to the destabilization of neuroplasticity pathways (including Brain-derived Neurotrophic Factor (BDNF-TrkB)) and reduce the adaptive capacity of the nervous system in PTSD and dysthymia. Research indicates that both acute and chronic stress activate the neuronal SUMOylation pathway, leading to increased levels of SUMO1 and enzymes catalyzing its attachment to proteins, such as E1 (SAE1/SAE2), E2 (UBC9), and E3 ligases [12]. This mechanism has a neuroprotective effect because it modifies transcription factors, cytoskeletal proteins, receptors, and enzymes involved in this process. However, some authors emphasize that activating SUMOylation may also result in maladaptive neuronal changes, which could prolong PTSD symptoms. SUMOylation can both activate and suppress the transcription of genes related to neuroplasticity, neurogenesis, and apoptosis [14]. In particular, modification of CREB and BDNF regulators is associated with reduced expression of the BDNF protein. Decreased serum BDNF levels can lead to hippocampal volume reduction, weakening of long-term synaptic potentiation (LTP), and cognitive and emotional deficits reflecting clinical depression symptoms [7,8,12]. Moreover, in response to oxidative stress, SUMO1affects the function of DNA repair proteins, such as p53 or Topoisomerase-1 (Topo-1), contributing to apoptosis activation. Nonetheless, chronic stress can also cause neuronal degeneration, especially in brain areas involved in emotional memory, like the amygdala, hippocampus, and septal nuclei. From a molecular perspective, in neurons, SUMO1 integrates processes activated in PTSD: oxidative stress induction, transcriptional disturbances, apoptosis activation, and neuroinflammation [15,16,17].

Another important factor involved in neuroinflammation in PTSD and affective disorders is MDA, an end product of membrane lipid peroxidation that is recognized as a stable and sensitive oxidative stress biomarker [18]. MDA plays a crucial role in neuronal dysregulation and the induction of central nervous system inflammation. MDA is mainly generated by reactive oxygen species (ROS) acting on polyunsaturated fatty acids in cell membranes, including neurons and glial cells [19]. Lipid peroxidation leads to fragmentation of membrane phospholipids and formation of cytotoxic aldehydes, such as MDA, 4-hydroxynonenal (4-HNE), and acrolein. Due to its electrophilic properties, MDA reacts with proteins, DNA, and phospholipids, forming stable adducts that disrupt cellular homeostasis. Elevated peripheral lipid peroxidation may indicate increased oxidative stress and an insufficient antioxidant defense in affective disorders and PTSD. Lipid-derived inflammatory and neurodegenerative mediators are important in PTSD and affective disorder pathophysiology, as they can alter pro-inflammatory cytokine regulation. Lipid peroxidation may generate immunogenic neoepitopes, which in turn activate autoimmune reactions associated with PTSD and dysthymia. These changes can disrupt affect regulation and trigger neurodegenerative processes [20,21,22,23]. Increased affective episodes may sensitize neurohormonal stress responses and ignite neuroinflammation (kindling). MDA may intensify stress in PTSD and progress dysthymic disorders into major depression through several mechanisms: inhibition of neurogenesis via BDNF-related signaling in the hippocampus, promotion of inflammation by microglial activation and increased pro-inflammatory cytokine production, mitochondrial dysfunction through respiratory chain damage, decreased ATP production, and enhanced ROS generation. Additionally, lipid peroxidation alters receptor function by modulating glutamatergic (including NMDA) receptors, triggering excitotoxicity, and MDA exerts epigenetic effects by reacting with histone proteins and DNA [18,19,20]. The next factor worth noting in the process of neuroinflammation is the chemokine CX3CL1, which is a cytokine protein belonging to the CX3C chemokine family. In the body, fractalkine performs a dual function—one of its forms is a membrane-bound protein on neurons, and the other, in its soluble form, interacts with the G protein-coupled receptor CX3CR1, which is mostly located in microglia within the amygdala, hippocampus, and prefrontal cortex. When this chemokine interacts with its receptor, proteolytic cleavage of the extracellular domain of CX3CL1 occurs (by metallopeptidase domain 10-ADAM-10, metallopeptidase domain 17-ADAM-17, or γ-secretase), releasing it into the extracellular space [24,25]. This phenomenon takes place under the influence of stress, neuronal injury, or the pro-inflammatory signals present in PTSD and dysthymia. The migration of pro-inflammatory cytokines into the CNS and the secondary activation of microglia can also occur in clinical situations associated with increased permeability of the blood–brain barrier [26]. The inflammatory response of microglia is maintained (potentiated) along the CX3CL1/CX3CR1 pathway through altered expression of genes related to this process, including factors such as Tumor Necrosis Factor-α (TNF-α), Interleukin-1 beta (IL-1β), and Interleukin-6 (IL-6). This specific ligand–receptor interaction makes the CX3CL1–CX3CR1 pathway a key regulatory mechanism in neuron–microglia communication and in maintaining neuroimmune balance. Microglia are a population of immune cells in the CNS that can adopt phenotypes with different functional characteristics: classically activated (M1, pro-inflammatory) or alternatively activated (M2, neuroprotective). CX3CL1 modulates this activity by promoting the transformation of microglia into the M2 phenotype, reducing the production of cytokines and nitric oxide (NO) while increasing the expression of trophic factors such as Insulin-like Growth Factor (IGF-1) and BDNF. Sustained fractalkine expression in neurons enables communication with microglia through the chemotactic and adhesive properties of this chemokine. Many authors have indicated that it plays an important role in regulating inflammation, synaptic plasticity, and neuroprotection [27,28,29,30]. Moreover, disturbances involving the activation of pro-inflammatory factors through the CX3CL1–CX3CR1 interaction pathway may lead to the ignition (kindling) of neuroinflammation, which is a predictor of microglial dysfunction. This can result in altered CX3CL1 expression and receptor CX3CR1 dysfunction, shifting microglial activity toward a pro-inflammatory state; the loss of organismal homeostasis; and neurohormonal disturbances of the HPA axis, which accompany PTSD and dysthymia [31,32]. Consequently, neurotoxicity may occur, leading to a loss of synaptic integrity, reduced neurogenesis in the hippocampus, and atrophy of fronto-limbic structures. In addition to its immunomodulatory function, fractalkine also plays an important role in synaptic plasticity and the synchronization of neuronal circuits.

Studies conducted in animal models have demonstrated that signaling along the CX3CL1–CX3CR1 pathway influences many important neurobiological processes, such as glutamatergic and GABAergic transmission, long-term potentiation in the hippocampus, maturation of dendritic spines, and regulation of the proliferation and differentiation of progenitor cells in the dentate gyrus region [32,33]. Furthermore, researchers have found that disturbances in these processes may contribute to the occurrence of cognitive impairments and anhedonia, symptoms that are characteristic of affective disorders such as dysthymia and of PTSD, which is classified as a trauma- and stress-related disorder [34].

Another important biochemical factor related to neuroinflammation in PTSD and affective states is UCHL1, which is also referred to as PGP9.5 in the literature. This enzyme belongs to the deubiquitinase (DUB) family and plays a key role in the ubiquitin–proteasome system (UPS). The gene encoding UCHL1 is located on chromosome 4p14 and is characterized by high neuronal expression. Ubiquitin Carboxy-terminal Hydrolase (UCH) recognizes and cleaves isopeptide bonds between the last amino acid of the ubiquitin molecule and other proteins. UCH removes unnecessary fragments and activates free ubiquitin molecules needed for the ubiquitination pathway to function. The second function of UCH is the ubiquitination of substrate proteins. This process occurs under the influence of ubiquitin ligase (E3), an enzyme responsible for specifically recognizing target (substrate) proteins to be modified with ubiquitin. This action is based on recognizing “protein quality” and selectively degrading unnecessary substances. This selection facilitates the maintenance of proper protein homeostasis, immune response modulation, cell cycle regulation, and synaptic plasticity. UCHL1 is usually located in the cytoplasm, but under pathological conditions, it can be found in the mitochondria or cell nucleus. Under oxidative stress, cysteine residues in this enzyme become oxidized, which can lead to enzymatic deactivation. UCHL1 is particularly sensitive to post-translational modifications such as nitrosylation or carbonylation [35], which can also alter its functions. UCHL1 also participates in the recycling of free ubiquitin, maintaining the balance between the free and conjugated forms of ubiquitin. Studies have shown that, in neurons, UCHL1 protects cells against the accumulation of degraded proteins. This is particularly important, as neuronal proteins with long half-lives are especially susceptible to damage [35,36]. Furthermore, publications by Ma A. et al. [37] and Kampuraj D. et al. [38] have shown that UCHL1 interacts with cytoskeletal filaments such as neurofilaments, suggesting that it participates in regulating neuronal morphology and plasticity. Dysfunction of UCHL1 under intense oxidative stress can lead to the accumulation of ubiquitinated proteins and neuronal cell death. Moreover, UCHL1 is referred to in the literature as a biomarker of mental disorders [34,35,36,37,38].

This article focuses on the assessment of four biomarkers and the evaluation of dysthymia in individuals with a history of PTSD (Past PTSD ≤ 5 years and Past PTSD > 5 years) compared to a control group with no history of PTSD. The biomarkers assessed in blood serum were SUMO1, MDA, CX3CL1 (fractalkine), and UCHL1, which represent complex cellular mechanisms related to proteostasis, lipid peroxidation, inflammatory signaling, and the cellular stress response.

The aim of this study was to evaluate the impact of PTSD and its chronicity on clinical profiles, biomarker levels (SUMO1, MDA, CX3CL1, and UCHL1), and CDRS scores in male patients, comparing Past PTSD ≤ 5 y (N = 33), Past PTSD > 5 y (N = 31), and No PTSD controls (N = 28) (Section 2.1, Section 2.2, Section 2.3).

This study also aimed to explore correlations between biomarkers and CDRS domains across groups, identifying neurobiological mechanisms and differences by PTSD status and duration (Section 2.4).

We hypothesize that alterations in the serum levels of SUMO1, MDA, CX3CL1, and UCHL1 are associated with the severity of dysthymic symptoms in individuals with a history of PTSD, reflecting the involvement of oxidative stress, neuroinflammatory, and proteostatic mechanisms in affective dysregulation.

## 2. Results

### 2.1. Characteristics of Demographic and Addiction-Related Profiles in PTSD

This study sample comprises 92 male participants aged 19 to 50 years, among whom 64 have a history of PTSD and 28 serve as controls with no history of PTSD. The PTSD group is further stratified by the time since PTSD diagnosis, using a clinically meaningful cutoff of 5 years: 33 participants have had PTSD for 5 years or less (Past PTSD ≤ 5 y), and 31 have had PTSD for more than 5 years (Past PTSD > 5 y). This stratification allows for an examination of the impact of PTSD chronicity on clinical outcomes, reflecting a well-established approach in psychiatric research to differentiate between recent and long-standing effects of trauma.

Detailed demographic and clinical characteristics of the participants, including their age distribution, engagement with psychotherapy, and self-destructive behaviors, are provided in Table 1.

Participants in the Past PTSD ≤ 5 years group demonstrated the highest rates of psychotherapy utilization and the longest treatment durations among all cohorts, predominantly involving cognitive behavioral therapy. Self-destructive behaviors were present in a minority, primarily manifesting as self-harm, with lesser occurrences of other forms and tattoos, and no restrictive eating habits noted.

In the Past PTSD > 5 years group, psychotherapy engagement remained high, though with shorter durations than the acute subgroup, and cognitive behavioral therapy was the main modality, supplemented by psychodynamic and other approaches. Self-destructive behaviors were markedly more common here, encompassing self-harm as the leading type, followed by tattoos, other behaviors, and a small proportion of restrictive eating habits—indicating potential escalation linked to prolonged PTSD.

The control group showed the lowest psychotherapy involvement, with brief durations and a mix of therapeutic types led by cognitive behavioral therapy. Self-destructive behaviors occurred at rates similar to the acute PTSD subgroup, chiefly self-harm, alongside tattoos and other forms, but without restrictive eating habits.

The Past PTSD ≤ 5 years and Past PTSD > 5 years groups do not differ significantly in psychotherapy duration (adjusted p = 0.190). However, the Past PTSD ≤ 5 years group exhibits a significantly longer duration compared to controls (adjusted p = 0.009), while no such difference is observed between the Past PTSD > 5 years group and controls (adjusted p = 0.090). This pattern may reflect greater initial treatment engagement in individuals with more recent PTSD diagnoses, potentially driven by acute symptom management needs, whereas chronic cases and controls show comparable lower durations.

Psychotherapy engagement is significantly more prevalent in both PTSD subgroups compared to controls (adjusted p = 0.001 for overall group comparison), with no difference between the Past PTSD ≤ 5 years and Past PTSD > 5 years groups (adjusted p = 0.765). This indicates a higher utilization of therapeutic interventions among those with a PTSD history, irrespective of chronicity.

Self-destructive behaviors are significantly more prevalent in the Past PTSD >5 years group compared to both the Past PTSD ≤ 5 years group and controls (adjusted p < 0.001 for both pairwise comparisons), indicating a stronger association with chronic PTSD, possibly attributable to prolonged cumulative stress, neurobiological adaptations such as hypothalamic–pituitary–adrenal axis dysregulation, or unresolved trauma effects. No significant difference is found between the Past PTSD ≤ 5 years group and controls (adjusted p = 1.000). Age, duration of hazardous employment, and type of therapy show no significant differences across groups.

### 2.2. Biomarker Profiles in Patients with Past PTSD and Controls

Oxidative stress and neuroinflammatory processes are implicated in the pathophysiology of post-traumatic stress disorder (PTSD), in which they potentially influence disease progression and symptom severity. Table 2 presents the levels of four biomarkers—SUMO1, MDA, CX3CL1, and UCHL1—in male patients with past PTSD (stratified by duration since diagnosis: ≤5 years and >5 years) and controls without PTSD, offering insights into the impact of PTSD and its chronicity on these parameters. The results across PTSD status groups are visualized in Figure 1.

For SUMO1, a protein involved in post-translational modification and stress response, the No PTSD group exhibits the highest median level at 34.63 ng/mL (IQR 26.26–41.76), followed by the Past PTSD > 5 y group at 11.40 ng/mL (IQR 6.67–19.59) and the Past PTSD ≤ 5 y group at 4.47 ng/mL (IQR 3.51–5.44). The post hoc result (a < b < c) indicates a progressive increase in SUMO1 levels from Past PTSD ≤ 5 y to Past PTSD > 5 y to No PTSD. This implies that PTSD is associated with reduced SUMO1 levels compared to the controls, potentially reflecting impaired stress response mechanisms in PTSD patients. The higher levels in the >5 y group compared to the ≤5 y group further indicate that a longer PTSD duration may partially mitigate this suppression, possibly due to adaptive cellular responses over time.

In contrast, MDA, a marker of oxidative stress, shows the opposite trend, with the highest median level observed in the Past PTSD ≤ 5 y group at 23.50 nmol/mL (IQR 18.66–30.71), followed by the Past PTSD > 5 y group at 8.37 nmol/mL (IQR 5.57–12.72) and the No PTSD group at 3.13 nmol/mL (IQR 2.28–3.65). The post hoc result (a > b > c) confirms that MDA levels are significantly elevated in the PTSD patients compared to the controls, with the highest levels measured in those with more recent PTSD (≤5 y). This infers that oxidative stress is a prominent feature of PTSD, particularly in the early years after diagnosis, possibly due to heightened reactive oxygen species production in response to acute trauma. The lower levels in the >5 y group compared to the ≤5 y group indicate that oxidative stress may decrease over time, potentially reflecting physiological adaptation or the effects of long-term coping mechanisms.

CX3CL1, a chemokine involved in neuroinflammation, follows a similar pattern to MDA, with the highest median level observed in the Past PTSD ≤ 5 y group at 30.13 ng/mL (IQR 22.66–38.27), followed by the Past PTSD > 5 y group at 8.60 ng/mL (IQR 6.02–15.69) and the No PTSD group at 3.59 ng/mL (IQR 3.09–4.96). The post hoc result (a > b > c) indicates that CX3CL1 levels are significantly elevated in the PTSD patients, particularly those with more recent PTSD, compared to the controls. This elevation indicates that neuroinflammation is a key feature of PTSD, likely driven by immune activation in response to trauma, with the most pronounced effect observed in the ≤5 y group. The reduction in CX3CL1 in the >5 y group compared to the ≤5 y group implies that inflammatory processes may subside over time, possibly due to the resolution of acute immune responses or the impact of therapeutic interventions.

UCHL1, a neuronal protein associated with protein degradation and neuroprotection, also shows higher levels in PTSD patients, with the Past PTSD ≤ 5 y group at 30.54 ng/mL (IQR 19.92–36.16), the Past PTSD > 5 y group at 11.18 ng/mL (IQR 6.38–16.69), and the No PTSD group at 3.98 ng/mL (IQR 3.04–4.67). The post hoc result (a > b > c) mirrors the trends for MDA and CX3CL1, indicating that UCHL1 levels are elevated in PTSD, with the highest levels observed in the ≤5 y group. This elevation may reflect neuronal stress or compensatory neuroprotective mechanisms in response to trauma, which are more pronounced in recent PTSD. The lower levels in the >5 y group indicate a potential decline in these mechanisms over time, possibly due to chronic neuronal adaptation or exhaustion.

Effect of PTSD on studied parameters

The presence of PTSD significantly affects all studied biomarkers, with distinct patterns observed in the PTSD patients compared to the controls. SUMO1 levels are reduced in the PTSD patients, demonstrating impaired stress response pathways, while MDA, CX3CL1, and UCHL1 levels are elevated, indicating heightened oxidative stress, neuroinflammation, and neuronal stress responses, respectively. These findings emphasize the role of PTSD in driving systemic and neurobiological changes, which may contribute to the persistence of symptoms and comorbidities such as anxiety or depression.

Effect of PTSD duration on patient profile

The duration of PTSD also significantly influences the patient profile within the studied parameters. For MDA, CX3CL1, and UCHL1, patients with more recent PTSD (≤5 y) exhibit the highest levels, implying that the early years following PTSD diagnosis are characterized by intense oxidative stress, neuroinflammation, and neuronal stress responses. In contrast, patients with longer-standing PTSD (>5 y) show reduced levels of these biomarkers, indicating a potential attenuation of these processes over time, possibly due to physiological adaptation, chronic coping mechanisms, or the cumulative effects of interventions such as psychotherapy (as noted in earlier sections). However, for SUMO1, the >5 y group has higher levels than the ≤5 y group, evincing that a longer PTSD duration may lead to a partial recovery of stress response mechanisms, though they remain below control levels.

These differences highlight the evolving nature of PTSD’s impact on biomarker profiles, with acute and chronic phases presenting distinct challenges that may require tailored therapeutic approaches. For example, early interventions targeting oxidative stress and inflammation (e.g., antioxidants and anti-inflammatory agents) may be more critical in the ≤5 y group, while strategies to enhance stress response pathways (e.g., SUMO1-related therapies) could be explored in chronic PTSD.

### 2.3. Cornell Dysthymia Rating Scale (CDRS) Results in Patients with Past PTSD and Controls

Table 3 presents the CDRS scores for 20 domains and the total score across the male patients with past PTSD (stratified by duration: ≤5 years and >5 years) and the controls without PTSD, highlighting significant differences that reflect the interplay between PTSD chronicity and dysthymic symptoms.

Characterization of patient profiles across CDRS domains

For the Past PTSD ≤ 5 y group, the mean domain scores range from 1.88 for insomnia (INS) to 2.97 for low energy (LOE), with a total CDRS score of 51.70 (SD 7.38). This group exhibits moderate symptom severity across most domains, with notable elevations in social withdrawal (SWD, 2.94), low attention/concentration (LAC, 2.94), and psychic anxiety (PSA, 2.94), evincing significant social and cognitive impairment alongside anxiety in the early years following PTSD diagnosis. The Past PTSD > 5 y group has higher mean scores across nearly all domains, ranging from 2.29 for SWD to 3.77 for low sexual interest/activity (LSA), with a total CDRS score of 65.03 (SD 6.89). This group demonstrates severe symptom severity, particularly in LSA (3.77), pessimism (PESS, 3.71), and depressed mood (DM, 3.68), indicating profound emotional and sexual dysfunction in chronic PTSD. The No PTSD group has the lowest scores, ranging from 0.00 for LOE to 1.82 for indecisiveness (IND), with a total CDRS score of 15.00 (SD 3.57), reflecting minimal to mild symptom severity, with most domains below 1.5 except for IND (1.82) and somatic anxiety (SOA, 1.54).

Across emotional domains (DM, PESS, Guilt-GUI), the Past PTSD > 5 y group consistently shows the highest scores (e.g., DM: 3.68, PESS: 3.71, GUI: 3.61), followed by the Past PTSD ≤ 5 y (e.g., DM: 2.00, PESS: 2.82, GUI: 2.79) and No PTSD (e.g., DM: 0.18, PESS: 1.00, GUI: 1.29) groups, with post hoc tests (a < b, c < a) confirming significant differences between all groups. The cognitive domains (low self-esteem (LSE), lack of control (LC), IND, and LAC) follow a similar pattern, with the Past PTSD > 5 y group having the highest scores (e.g., LSE: 3.39, LC: 2.87), the Past PTSD ≤ 5 y having intermediate scores (e.g., LSE: 2.12, LC: 2.30), and the No PTSD group scoring the lowest (e.g., LSE: 1.39, LC: 0.43), though no significand differences were found for IND and LAC between the PTSD groups (a, a, b). The somatic domains (PSA, SOA, somatic general (SOG), low productivity (LOP), LOE, LSA, INS, and DIV) also reflect this trend, with the Past PTSD > 5 y group exhibiting severe symptoms (e.g., PSA: 3.61, LSA: 3.77, INS: 3.42), the Past PTSD ≤ 5 y group exhibiting moderate symptoms (e.g., PSA: 2.94, LSA: 2.55, INS: 1.88), and the No PTSD group exhibiting minimal symptoms (e.g., PSA: 0.86, LSA: 0.57, INS: 0.43). There were exceptions like LOE and DIV, for which the PTSD groups do not differ (a, a, b). The social and behavioral domains (SWD, IEA, WOR) show that the Past PTSD > 5 y group scores the highest (e.g., IEA: 3.45, WOR: 3.55), the Past PTSD ≤ 5 y group has intermediate scores (e.g., IEA: 2.82, WOR: 2.64), and the No PTSD group scores the lowest (e.g., IEA: 0.93, WOR: 1.43), with no difference between the PTSD groups for SWD (a, a, b). Suicidal ideation (SUI) is elevated in both PTSD groups (Past PTSD ≤ 5 y: 2.48, Past PTSD > 5 y: 2.81) compared to the control group (0.18), with no significant difference between the PTSD groups (a, a, b). See Figure 2.

Effect of PTSD on studied CDRS domains

PTSD significantly affects CDRS scores and the total score across all domains, as evidenced by one-way ANOVA (*p* < 0.001) and post hoc Tukey’s HSD tests. Both PTSD groups exhibit substantially higher scores than the No PTSD group in every domain, with total scores of 51.70 (Past PTSD ≤ 5 y) and 65.03 (Past PTSD > 5 y) compared to 15.00 (No PTSD). The effect sizes are large; for the total score, Cohen’s d for Past PTSD ≤ 5 y vs. No PTSD is 6.24, and it is 8.92 for Past PTSD > 5 y vs. No PTSD, indicating a profound impact of PTSD on dysthymic symptom severity. This entails that PTSD exacerbates emotional (e.g., depression, guilt), cognitive (e.g., low self-esteem, indecision), somatic (e.g., anxiety, low energy), and social (e.g., withdrawal, irritability) symptoms, likely reflecting the neurobiological overlap between PTSD and dysthymia, such as dysregulation of the hypothalamic–pituitary–adrenal axis and heightened amygdala activity.

Based on CDRS severity categories, the No PTSD group (15.00) falls within the healthy range (≤40), indicating minimal dysthymic symptoms. The Past PTSD ≤ 5 y group (51.70) scores within the range of mild dysthymia (41–53 for overall score), while the Past PTSD > 5 y group (65.03) scores within the range of moderate dysthymia (54–67), approaching the severe threshold (68–80). The effect sizes are substantial; Cohen’s d for Past PTSD ≤ 5 y vs. No PTSD is d = 6.24, and it is 8.92 for Past PTSD > 5 y vs. No PTSD, highlighting a profound impact of PTSD on dysthymic symptom severity.

These results indicate that PTSD markedly exacerbates dysthymia, with PTSD patients exhibiting mild to moderate symptom severity compared to the healthy profile of controls.

Effect of PTSD duration on studied CDRS domains

The duration of PTSD significantly differentiates the profiles of patients with PTSD across most CDRS domains, with Past PTSD > 5 y patients exhibiting more severe symptoms than Past PTSD ≤ 5 y patients. The total CDRS score is higher in the >5 y group (65.03 vs. 51.70, *p* < 0.05, Cohen’s d = 1.85), a large effect indicating a greater overall dysthymic burden in chronic PTSD. In 14 of 20 domains (DM, lack of interest or pleasure (LIP), PESS, LSE, GUI, LC, PSA, SOA, WOR, IEA, SOG, LOP, LSA, and INS), the Past PTSD > 5 y group’s scores are significantly higher than the Past PTSD ≤ 5 y group’s scores (e.g., PESS: 3.71 vs. 2.82, LSA: 3.77 vs. 2.55), with post hoc results (a < b) confirming these differences. Exceptions include SUI, SWD, IND, LAC, LOE, and DIV, where the PTSD groups do not differ significantly (a, a), though the scores for both groups remain elevated compared to the No PTSD group’s scores (b). This indicates that longer PTSD duration is associated with worsening emotional (e.g., pessimism, guilt), cognitive (e.g., low self-esteem), somatic (e.g., low sexual interest, insomnia), and social (e.g., irritability) symptoms, potentially due to cumulative stress, maladaptive coping, or progressive neurobiological changes. Clinically, these findings underscore the need for intensified interventions targeting chronic PTSD patients, focusing on severe emotional and somatic symptoms in particular, while early interventions in the ≤5 y group may mitigate progression to this more severe profile.

Applying the CDRS severity categories, the Past PTSD ≤ 5 y group (51.70) scores within the range of mild dysthymia (41–53), while the Past PTSD > 5 y group (65.03) scores within the range of moderate dysthymia (54–67), approaching the severe threshold (68–80). The effect size for this difference is large, with Cohen’s d = 1.85, indicating a substantial increase in symptom severity with longer PTSD duration. This evinces that chronic PTSD (>5 y) is associated with a more severe dysthymic profile, potentially due to cumulative stress, entrenched maladaptive coping, or progressive neurobiological changes.

### 2.4. Analysis of Correlations Between Biomarkers and CDRS Domains in Patients with Past PTSD and Controls

The relationship between biomarkers (SUMO1, MDA, CX3CL1, and UCHL1) and CDRS domains provides insights into the neurobiological underpinnings of PTSD and its chronicity, potentially guiding targeted interventions. Table 4 presents the Spearman correlation coefficients (ρ) between these biomarkers and the CDRS domains across three groups studied, highlighting significant associations that differ by PTSD status and duration.

In the Past PTSD ≤ 5 y group, two significant correlations are observed: UCHL1 with LIP (ρ = −0.36, *p* = 0.042) and UCHL1 with DIV (ρ =0.45, *p* = 0.009), which indicate that higher UCHL1 levels are linked to reduced anhedonia but increased diurnal symptom variation in recent PTSD. In the Past PTSD > 5 y group, three significant correlations emerge: SUMO1 with LIP (ρ = −0.41, *p* = 0.023), CX3CL1 with LIP (ρ = −0.40, *p* = 0.026), and CX3CL1 with DIV (0.41, *p* = 0.021), which demonstrate that elevated SUMO1 and CX3CL1 levels are associated with decreased interest/pleasure, while CX3CL1 correlates positively with diurnal variation in chronic PTSD. The No PTSD group shows two significant correlations: SUMO1 with LIP (ρ = −0.39, *p* = 0.038) and UCHL1 with LC (ρ = −0.38, *p* = 0.044). These correlations, which are both negative, imply that in the controls, higher SUMO1 levels reduce anhedonia, and higher UCHL1 levels lessen the sense of a lack of control. Compared to the No PTSD group (with two significant correlations), the PTSD groups exhibit more frequent associations (2 in ≤5 y, 3 in >5 y), focusing on LIP (anhedonia) and DIV (diurnal variation) in PTSD patients versus LIP and LC in the controls, highlighting PTSD’s role in shaping symptom–biomarker relationships. The PTSD groups share LIP and DIV as key domains, but their biomarkers differ: ≤5 y involves UCHL1, while >5 y involves SUMO1 and CX3CL1, indicating a shift in biomarker interactions over time. The broader biomarker associations with LIP in the >5 y group demonstrate more diverse neurobiological interactions in chronic PTSD, potentially involving inflammatory (CX3CL1) and protein regulation (SUMO1) pathways, whereas UCHL1 predominates in the ≤5 y group, possibly due to neuronal stress responses. Clinically, these findings indicate that early PTSD interventions might target UCHL1-related pathways to mitigate anhedonia and diurnal variation, while patients with chronic PTSD may benefit from strategies addressing SUMO1 and CX3CL1 to manage persistent anhedonia, potentially through anti-inflammatory or neuroprotective approaches.

These correlations suggest potential links but require further investigation to determine directionality. We fully agree that cross-sectional studies, by their nature, are unable to determine temporal precedence or rule out reverse causality, and can only identify associations or correlations. This is a well-recognized constraint in observational research, particularly in psychiatric and neurobiological investigations where dynamic processes like stress responses and symptom progression unfold over time.

## 3. Discussion

Post-traumatic stress disorder arises from exposure to trauma and manifests as intrusion symptoms, avoidance, negative alterations in cognitions and mood, and alterations in arousal and reactivity. Increasing evidence indicates that PTSD can co-occur with affective disorders such as dysthymia. In the DSM-5, dysthymia is now referred to as Persistent Depressive Disorder (PDD). To be diagnosed with PDD, an individual must experience a depressed mood for most of the day, more days than not, for at least two years. Additionally, at least two of the following symptoms must be present: poor appetite or overeating, insomnia or hypersomnia, low energy or fatigue, low self-esteem, poor concentration or difficulty making decisions, and feelings of hopelessness [1,2,3].

In recent years, increasing attention has been paid to neurobiological biomarkers that may serve as objective indicators of the pathophysiological processes underlying these disorders. Potential biomarkers include SUMO1, MDA, CX3CL1, and UCHL1. SUMO1 participates in post-translational protein modifications under oxidative stress and may reflect adaptive neuronal mechanisms in response to chronic stress [16,18,25,27,28,37]. MDA is an indicator of lipid peroxidation and cellular membrane damage caused by oxidative stress, correlating with the severity of depressive and PTSD symptoms [18,19]. CX3CL1 is a chemokine involved in neuron–microglia signaling. It plays a key role in the brain’s neuroinflammatory response. Dysregulation Dysregulated CX3CL1 expression may contribute to the persistence of mood disorders [27,28,32]. UCHL1 is involved in regulating the ubiquitin–proteasome pathway, which maintains neuronal protein homeostasis. Changes in UCHL1 expression are observed in both neurodegenerative diseases and affective disorders [36,37].

In this study, we assessed the impact of PTSD chronicity on patients’ clinical presentation and biochemical biomarker levels (SUMO1, MDA, CX3CL1, and UCHL1), aiming to distinguish between relatively recent PTSD (symptom duration ≤ 5 years) and chronic PTSD (>5 years). Specifically, we found that in patients with PTSD lasting ≤5 years, self-destructive behaviors occurred in 39.4% of subjects. Among these behaviors, self-harm was predominant (69.2%), followed by other self-destructive behaviors (23.1%) and tattoos (7.7%). In contrast, in the group with PTSD lasting >5 years, the frequency of self-destructive behaviors was significantly higher at 83.9%. In this group, self-harm was observed in 53.8% of patients, tattoos in 19.2%, other self-destructive behaviors in 15.4%, and restrictive eating habits in 11.5%. In the context of PTSD, these behaviors may be interpreted as dysfunctional emotion regulation strategies aimed at reducing intense affective symptoms, regaining a sense of control, or fulfilling a need for self-punishment [2]. Given the chronic health consequences of trauma, all patients received psychotherapy.

In our study, we found that plasma levels of the SUMO1 marker were the highest in the control group without PTSD (median 34.63 ng/mL), lower in the chronic PTSD group (>5 years; median 11.40 ng/mL), and the lowest in the recent PTSD group (≤5 years; median 4.47 ng/mL). These results indicate a significant relationship between SUMO1 levels and the time elapsed since the onset of PTSD symptoms. The observed reduction in SUMO1 levels, particularly in the group with more recent PTSD, may suggest that processes involving SUMOylation undergo lasting changes in response to traumatic stress, with intensity decreasing over time. Reduced SUMO1 levels may contribute to neuronal dysfunction, increased susceptibility to oxidative damage, impaired neuroplasticity, and reduced neuronal protection in brain regions responsible for emotion regulation, such as the hippocampus, amygdala, and prefrontal cortex. As the literature lacks studies on SUMO1 levels in PTSD and affective disorders, these analyses are pioneering research. However, preclinical studies in animal models have highlighted the important role of SUMO1 in stress mechanisms [13,14,15]. For example, Lin H.Y. et al. examined the effects of chronic social defeat stress on hippocampal proteome changes in mice, focusing on the ubiquitin–proteasome pathway and post-translational protein modifications like SUMOylation [39]. Their results showed a significant reduction in SUMO1 levels in stress-susceptible individuals, which they interpreted as a potential factor differentiating stress resistance and susceptibility. Wang L. et al. confirmed the significance of functional SUMOylation (including SUMO1–SUMO3) in proper episodic and emotional memory processing, with disruptions leading to anxiety-like behaviors and memory deficits [40]. These findings underscore SUMOylation as a key mechanism regulating neuronal plasticity and the integration of emotional–cognitive processes. SUMOylation, as an important post-translational modification, affects protein conformation, subcellular localization, and participation in signaling pathways and seems particularly sensitive to chronic dysregulation of the HPA axis. Our findings of higher SUMO1 levels in chronic PTSD patients compared to those with more recent PTSD may indicate activation of secondary adaptive mechanisms. These mechanisms, developing at the cellular level, may partially compensate for the initial suppression of SUMOylation pathways and attempt to restore molecular homeostasis under chronic stress conditions [16,17,41].

MDA levels showed a trend opposite to the other markers, with the highest levels observed in the recent PTSD group (≤5 years; median 23.50 nmol/mL), lower levels in the chronic PTSD group (>5 years; median 8.37 nmol/mL), and the lowest levels in the control group (median 3.13 nmol/mL). This may reflect long-term adaptive mechanisms or neurobiological compensatory processes related to oxidative–antioxidative balance in the central nervous system (CNS). Hasan H.M. et al. [42] demonstrated significantly elevated oxidative stress markers, including MDA, in women with PTSD, indicating cellular damage induced by chronic oxidative stress. Ogłodek E. [18] documented that depressive symptom severity in patients with or without PTSD correlated with MDA levels. Petrovic R. et al. [43] found significantly increased MDA concentrations in the amygdala and hippocampus of rats subjected to stress, highlighting the role of oxidative stress in the neurodegeneration of structures essential for emotion regulation.

A similar pattern of concentration changes was observed for the chemokine CX3CL1, which plays an important role in neuroinflammatory mechanisms. The highest median levels were recorded in the recent PTSD group (≤5 years; 30.13 ng/mL), with lower levels observed in the chronic PTSD group (>5 years; 8.60 ng/mL) and the lowest levels found in the control group (3.59 ng/mL). CX3CL1 is a neuron-specific chemokine that signals via the CX3CR1 receptor on microglia. High CX3CL1 levels may limit excessive microglial activation and counteract the development of neuroinflammatory responses after stress exposure. Goshi N. et al. showed that individuals with higher CX3CL1 levels after trauma had a lower risk of developing PTSD, suggesting that this chemokine could be used as a resilience marker [44]. Other studies confirmed that the CX3CL1–CX3CR1 system regulates synaptic plasticity, cytokine release, the stress response, cognitive processes, and behavior [45,46]. Variations in the activity of this pathway may therefore contribute to individual differences in stress vulnerability, microglial morphology, and neuronal plasticity.

In this study, we showed that UCHL1 levels are significantly elevated in PTSD patients compared to healthy controls. Analysis by PTSD duration revealed a clear trend: the highest values were seen in the recent PTSD group (≤5 years; median 30.54 ng/mL), while in the individuals with chronic PTSD (>5 years),UCHL1 levels decreased to a median of 11.18 ng/mL, and the control group had the lowest levels (median: 3.98 ng/mL). These results suggest increased activation of compensatory mechanisms in early PTSD, including heightened ubiquitin–proteasome system activity. Over time, UCHL1 levels decrease, possibly reflecting progressive neuronal degeneration and reduced capacity for defensive responses to oxidative stress [35]. UCHL1 plays a key role in neuronal function, completing the ubiquitination process by removing ubiquitin from proteins marked for degradation, enabling signaling protein recycling and maintaining proteasomal efficiency [36]. In PTSD, disruptions in UCHL1 function can lead to the accumulation of damaged proteins and the activation of cell death pathways [37]. Our clinical results support these hypotheses, highlighting UCHL1 as a potential biomarker of the active phase of PTSD and a possible indicator of neurotoxicity. Preclinical animal studies corroborate this: in PTSD models, especially those involving chronic social defeat stress (CSDS), significant changes in UCHL1 expression were observed in brain structures related to emotion regulation and stress response, particularly in the hippocampus and prefrontal cortex [37,47]. Lin H.Y. et al. [39] showed that chronic social stress led to dysregulation of the ubiquitination pathway and reduced UCHL1 levels, which correlate with behavioral PTSD symptoms like social withdrawal, anxiety, and working memory deficits. Decreased UCHL1 may result from prolonged stress exposure, leading to enzymatic resource depletion and the initiation of neurodegenerative processes. Beyond its enzymatic function, UCHL1 also provides neuroprotection, maintaining cytoskeletal integrity, regulating axonal transport, and preventing pathological protein aggregation. UCHL1 deficiency, as suggested by our data and experimental studies, may lead to dysfunction in these processes, contributing to pathology in PTSD and other neurodegenerative disorders [35,36].

From a neurobiological perspective, both PTSD and dysthymia are associated with dysfunction of limbic structures—especially the amygdala, hippocampus, and the anterior part of the cingulate gyrus cortex—which play a key role in emotion processing, mood regulation, and stress response. In clinical practice, standard PTSD assessment tools may not be sufficient to identify co-occurring dysthymia, the symptoms of which can be nonspecific. Individuals with PTSD may experience chronic low mood, which—without appropriate tools—can be mistakenly interpreted as a symptom of PTSD itself rather than a separate affective disorder. The coexistence of PTSD and dysthymia can worsen the patient’s health prognosis. Moreover, comorbidity may indicate the need for a more in-depth clinical analysis and personalized pharmacological and psychotherapeutic treatment, especially in individuals exhibiting higher levels of anhedonia, feelings of emotional emptiness, and greater difficulties in social adaptation [48]. Considering the importance of accurate clinical diagnosis, we used the CDRS to identify symptoms not captured by typical PTSD scales. Our results clearly indicate that past PTSD significantly affects the severity of dysthymic symptoms, with the duration of the disorder differentiating the depth of the clinical picture. Patients with PTSD lasting over five years exhibit the highest symptom intensity, reaching a level corresponding to moderate dysthymia, with a tendency toward severe dysthymia. The most intense symptoms occurred in the emotional (pessimism, depressed mood, and guilt), cognitive (low self-esteem, and poor concentration), somatic domains (low energy, insomnia, anxiety), and social (irritability and social withdrawal) domains. Our findings support the concept that PTSD, especially chronic PTSD, leads to an accumulation of emotional and cognitive deficits, which fosters the development of persistent depressive changes. The observed coexistence of dysthymic symptoms and neurobiological changes confirms the hypothesis of partially shared pathophysiological mechanisms of PTSD and depressive disorders. Dysregulation of the HPA axis, overactivity of the amygdala, chronic neuroinflammation, and oxidative stress are potential convergence points of these disorders. These results suggest that PTSD not only increases the risk of dysthymic symptoms but also deepens and consolidates them over time, leading to progression from mild to moderate and severe affective disorders [49].

The cross-sectional design precludes inferences about temporality or causality between biomarker alterations and symptom changes. Reverse causality—where dysthymic symptoms might influence biomarker levels—cannot be excluded, and future longitudinal studies are essential to clarify these relationships. Additionally, given the exploratory nature of our study, the relatively small sample size (N = 92), and the focus on targeted biomarkers rather than high-throughput omics data, bioinformatics approaches such as causal inference modeling or Mendelian randomization were not feasible at this stage, as they would be underpowered. Future studies could incorporate bioinformatics tools, such as causal network modeling or Mendelian randomization, to explore potential causal pathways, provided larger datasets with genetic or multi-omics components are utilized.

The duration of PTSD significantly affects the severity of dysthymia. Clinical analysis using the CDRS showed that individuals with PTSD lasting over 5 years present a significantly more severe symptom profile than patients with PTSD lasting up to 5 years. The mean total CDRS score was significantly higher in the chronic PTSD group (65.03) compared to the shorter-duration group (51.70). This corresponds to a transition from mild to moderate dysthymia and suggests a clear deterioration in mental functioning over time. Significant differences were observed in 14 of 20 CDRS domains covering emotional (e.g., pessimism, guilt), cognitive (low self-esteem), somatic (low libido, insomnia), and social (irritability) factors. This indicates that chronic PTSD is associated with a gradual worsening of symptoms, potentially as a result of accumulated stress, entrenched maladaptive coping mechanisms, or neurobiological changes [32]. A correlation analysis between biomarkers and CDRS domains revealed specific, PTSD-duration-dependent patterns. In the PTSD ≤ 5 year group, significant correlations involved the UCHL1 protein: higher levels were associated with less severe anhedonia (LIP domain) and greater daily symptom variability (DIV). In chronic PTSD (>5 years), new associations emerged: higher SUMO1 and CX3CL1 levels were linked to increased severity of anhedonia, and CX3CL1 was positively correlated with symptom variability throughout the day. In the control group, only two correlations were observed: a negative correlation between SUMO1 and LIP and a correlation between UCHL1 and the feeling of a lack of control. This may indicate a progressive shift in dominant molecular mechanisms from compensatory early neuronal responses to chronic inflammatory–degenerative neurobiological changes [49,50,51,52].

Our study was intentionally designed as an exploratory analysis to investigate potential correlations between these biomarkers and CDRS dimensions in the context of PTSD chronicity. This approach allowed us to generate hypotheses about underlying neurobiological mechanisms without implying causation, as emphasized in our Section 2 and Section 3. To establish causality, a larger, longitudinal study with randomization (e.g., randomized controlled trials or prospective cohort designs) would indeed be necessary. Such designs could track biomarker changes over time relative to symptom onset and progression, while controlling for confounders like treatment effects or comorbidities [53,54].

Several methodological limitations should be noted. The study included only male participants, which restricts the generalizability of the findings to female patients and individuals of other gender identities. This approach was used to reduce hormonal and neuroendocrine variability that could affect oxidative stress, inflammation, and SUMOylation processes (SUMO1, MDA, CX3CL1, UCHL1). However, given that PTSD is more common among women and may involve sex-specific neurobiological mechanisms, future studies should include both female participants and individuals of diverse gender identities. Moreover, biomarker analyses were based solely on serum samples. Cerebrospinal fluid and peripheral blood mononuclear cells were not examined, although they could provide more direct insight into central nervous system alterations. Therefore, the current findings should be viewed as peripheral indicators of central processes. Future studies combining serum, CSF, and PBMC analyses are warranted to better determine the biological origin and clinical significance of these molecular changes.

## 4. Materials and Methods

### 4.1. Participant Characteristics

This study included participants divided into two clinical groups and one control group, with each group consisting of 30 individuals. The clinical groups consisted of male individuals diagnosed with post-traumatic stress disorder (PTSD): one group with a PTSD duration of up to 5 years following the traumatic event, and another with PTSD persisting for more than 5 years. The control group comprised healthy male participants without a history of PTSD, who were matched in age and occupational exposure.

All participants were qualified for participation in this study by a board-certified psychiatrist and family medicine specialist, who is a co-author of this article. The inclusion criteria for the PTSD groups were male gender, aged between 18 and 50, current employment in an occupation involving exposure to extreme stress, and a clinical diagnosis of PTSD based on DSM-5 criteria, confirmed using the Clinician-Administered PTSD Scale for DSM-5 (CAPS-5).

The exclusion criteria applied to all groups included a current or past diagnosis of psychiatric or somatic illnesses, ongoing pharmacological treatment, substance or nicotine dependence (including medications and illicit drugs), legal incapacitation, or employment in uniformed services, such as the military or police.

This careful selection of participants was intended to ensure a homogeneous study population and minimize confounding variables.

### 4.2. Cornell Dysthymia Rating Scale (CDRS)

The Cornell Dysthymia Rating Scale is a tool developed to assess the severity of symptoms of dysthymia—Persistent Depressive Disorder. It consists of 20 items evaluated by a clinician, each of which is rated on a five-point scale ranking symptom severity: 0 (absent), 1 (slight), 2 (mild), 3 (moderate), and 4 (severe). The total score can range from 0 to 80 points. The higher the score, the greater the symptom severity.

The scale covers a wide range of symptoms characteristic of chronic low mood. The first item, depressed mood, refers to a dominant depressed mood. The second, lack of interest or pleasure, measures anhedonia, i.e., a loss of interest or pleasure in daily activities. Pessimism reflects negative expectations about the future, and suicidal ideation relates to the presence of suicidal thoughts.

The following items are low self-esteem, excessive guilt, a subjective feeling of a lack of control over one’s life, and avoidance of social contacts. The ninth item, IND, refers to difficulty in making decisions, while LAC relates to a reduced ability to concentrate and pay attention.

Other symptoms include psychic anxiety and somatic anxiety, indicating psychological anxiety and somatic anxiety, respectively. Worry measures the tendency toward chronic worrying, while irritability or excessive anger relates to irritability or inappropriate anger.

The fifteenth item, somatic general, includes nonspecific somatic symptoms such as pain or physical discomfort without a clear cause. Low productivity and low energy assess decreased productivity and chronic fatigue, respectively. Low sexual interest/activity refers to decreased libido and sexual activity. The penultimate item, insomnia, measures difficulties with sleep, and the last one, diurnal variation, relates to daily mood fluctuations characteristic of depression.

The interpretation of results is as follows: a score of 0–40 suggests no clinically significant symptoms, a score of 41–53 indicates mild dysthymia, a score of 54–67 indicates moderate dysthymia, and a score of 68–80 indicates severe dysthymia.

CDRS is a useful tool for assessing long-term depressive symptoms, especially in the context of diagnosing dysthymia and monitoring treatment [33,34].

### 4.3. Blood Sampling

At the time of participant enrollment, the peripheral serum concentrations of glutamine, glutathione, caspase-1, and NMDA receptor proteins were determined using single-point quantification via enzyme-linked immunosorbent assay (ELISA), employing commercially available kits and following the manufacturer’s standardized protocol. This method is based on a sandwich-type immunoassay format utilizing high-affinity monoclonal anti-human antibodies specific to each analyte. Venous blood samples were collected through routine procedures and processed by centrifugation to isolate serum. Aliquots were immediately stored at −80 °C to maintain biochemical stability and prevent proteolytic or cytokine degradation. Prior to analysis, frozen serum samples were thawed at 4 °C, gently vortexed to ensure homogeneity, and diluted 1:3 with the supplied dilution buffer to achieve analyte concentrations within the validated dynamic detection range of the assays.

### 4.4. Biomarker Analysis Procedure

The target biomarkers were quantified in the human serum samples using commercially available sandwich ELISA kits, in accordance with the manufacturers’ protocols. Each assay included a full set of recombinant standards for calibration, blank controls, and appropriately diluted serum samples. All samples were applied in duplicate to 96-well microtiter plates pre-coated with capture antibodies specific to each analyte of interest.

Following sample application, the plates were incubated at room temperature for 60 min with continuous orbital shaking at 300 rpm to ensure optimal antigen–antibody binding. After the initial incubation, the wells were washed thoroughly to remove unbound constituents, and a biotin-conjugated detection antibody was added. The plates were incubated under identical conditions for an additional 60 min. Following a second washing step, horseradish peroxidase (HRP)-conjugated streptavidin was introduced and incubated for 30 min at room temperature.

After a final wash to eliminate excess conjugate, 100 μL of tetramethylbenzidine (TMB) substrate solution was dispensed into each well and incubated for 10 min in the dark at room temperature. The enzymatic reaction was terminated by the addition of 50 μL of acidic stop solution. Absorbance was immediately measured at 450 nm using a calibrated microplate spectrophotometer E-Liza Mat 3000: Awareness Technology, Palm City, FL, USA.

Assay Sensitivity and Standardization

Standard curves were generated for each biomarker based on serial dilutions of recombinant protein standards supplied with the kits. The assay detection ranges, lower limits of quantification, and catalog numbers for all reagents are provided below to ensure reproducibility and methodological transparency.

SUMO1: Assay range, 0.15–40 ng/mL; sensitivity, 0.128 ng/mL; (Shanghai, China); and catalogue No. 201-12-5318.

MDA: Assay range, 0.75–100 nmol/mL; sensitivity, 0.515 nmol/mL; (Shanghai, China); and catalogue No. 201-12-5380.

CX3CL1: Assay range, 0.2–30 ng/mL; sensitivity, 0.102 ng/mL; (Shanghai, China); and catalogue No. 201-12-2102.

UCHL1: Assay range, 0.2–30 ng/L; sensitivity, 0.125 ng/mL; (Shanghai, China); and catalogue No. 201-12-2329.

### 4.5. Statistical Analysis

All statistical analyses were performed using R (version 4.3.1). Descriptive statistics were reported as medians with interquartile ranges (IQRs) for continuous variables with non-normal distributions (e.g., age, biomarker levels) and as means with standard deviations (SDs) for CDRS scores to detect subtle differences in symptom severity between groups. Categorical variables were presented as counts and percentages (n, %). Normality of continuous variables was assessed using the Shapiro–Wilk test within each group (Past PTSD ≤ 5 y, Past PTSD > 5 y, and No PTSD).

Group comparisons for continuous variables were conducted using appropriate statistical tests based on data distribution. For non-normally distributed variables (e.g., age, biomarker levels), the Kruskal–Wallis rank sum test was used, followed by post hoc pairwise comparisons with the Dunn test and Holm-Bonferroni correction to control for multiple comparisons (adjusted *p* < 0.05). For the CDRS domain scores, which were assumed to approximate normality, one-way analysis of variance (ANOVA) was employed, followed by Tukey’s Honest Significant Difference (HSD) test for post hoc pairwise comparisons (*p* < 0.05). Effect sizes for significant group differences were calculated: Cohen’s d was used for ANOVA comparisons (e.g., CDRS total score differences), and the Wilcoxon rank-test (r) effect size was used for variables with non-normal distribution.

Categorical variables were compared using Pearson’s Chi-squared test or Fisher’s exact test when the expected cell counts were less than 5 (e.g., types of psychotherapy, types of self-destructive behaviors). Host-hoc analysis was conducted using Peardon’ Chi squared test with FDR correction for multiple comparisons.

Correlations between biomarkers and CDRS domains were assessed using Spearman’s rank correlation coefficient (ρ) due to the non-normal distribution of the biomarkers and the ordinal nature of CDRS scores (0–4 scale). Correlations were calculated separately for each group, resulting in 80 correlations per group (4 biomarkers × 20 domains). Significance was determined using unadjusted *p*-values. To account for multiple comparisons, a Bonferroni-adjusted significance threshold was applied (*p* < 0.000625, 0.05/80 per group). Fisher’s z-transformation was used to compare correlation coefficients between groups, with two-tailed *p*-values reported.

Statistical significance for all tests was set at *p* < 0.05, unless otherwise adjusted for multiple comparisons. Effect sizes were reported to provide clinical context: Cohen’s d values were interpreted as small (0.2), medium (0.5), or large (0.8), and Spearman’s ρ values were interpreted as weak (0.1–0.3), moderate (0.3–0.5), or strong (>0.5). Missing data were handled by reporting reduced sample sizes where applicable (e.g., psychotherapy: N = 59; self-destructive behaviors: N = 50). All *p*-values are two-tailed.

## 5. Conclusions

This study confirms that PTSD has a long-lasting and dynamic impact on neurobiological functioning and affective symptoms, including the severity of dysthymia. Changes identified in the expression of biomarkers such as SUMO1 and UCHL1 indicate a phase-dependent nature of the molecular mechanisms accompanying PTSD, highlighting the necessity of an individualized therapeutic approach.

SUMO1 emerges as a potential biomarker of susceptibility to PTSD, PTSD duration, and PTSD remission, as well as an indicator of the effectiveness of therapies aimed at restoring neuroplasticity. The use of sumoylation modulators is a promising therapeutic direction that integrates perspectives from epigenetics, neurochemistry, and molecular psychiatry, though further research is required.

UCHL1 may play a dual role in PTSD as a marker of neuronal damage and a potential target for neuroprotective interventions. Research into the pharmacological or biological modulation of UCHL1 activity may open new avenues for treating the neurodegenerative consequences of PTSD.

These conclusions emphasize the need to consider the duration of PTSD as a key factor differentiating patients’ symptoms and molecular profiles. In addition, therapy should be tailored to the phase of the disorder: in the acute phase, the priority is to reduce oxidative stress and neuroinflammation, whereas in the chronic phase, the focus should be on supporting neuroregeneration and adaptive post-translational mechanisms.

In summary, our results reinforce the call for the development of long-term, individualized therapeutic strategies and further investigation of the molecular pathways involved in PTSD.

Longitudinal, randomized studies with larger cohorts are warranted to assess temporality, causality, and the potential of these biomarkers as predictive tools for PTSD-related affective dysregulation.

## Figures and Tables

**Figure 1 ijms-26-10214-f001:**
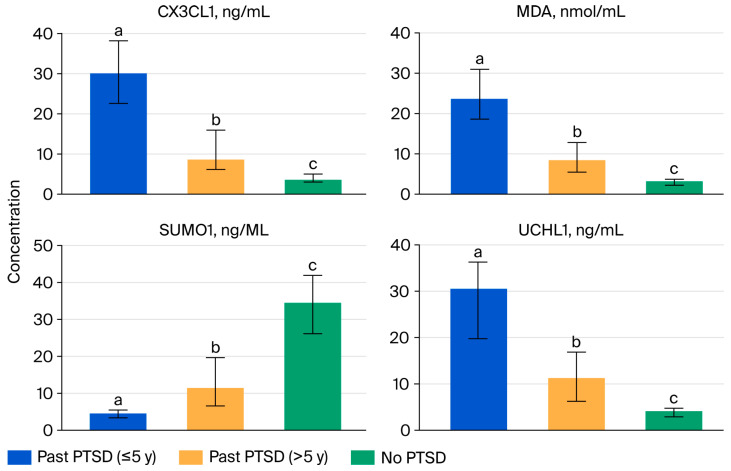
Biomarker levels (median and interquartile range, IQR) across PTSD status groups. Notes: Data are presented as median (IQR) values. *p*-values were calculated using the Krus-kal–Wallis rank sum test, with *p* < 0.001 indicating significant differences across groups for all biomarkers. Post hoc pairwise comparisons were performed using the Dunn test with Bonferroni correction; groups with different superscript letters (a, b, c) differ significantly (*p* < 0.05), while those sharing the same letter do not. PTSD = post-traumatic stress disorder.

**Figure 2 ijms-26-10214-f002:**
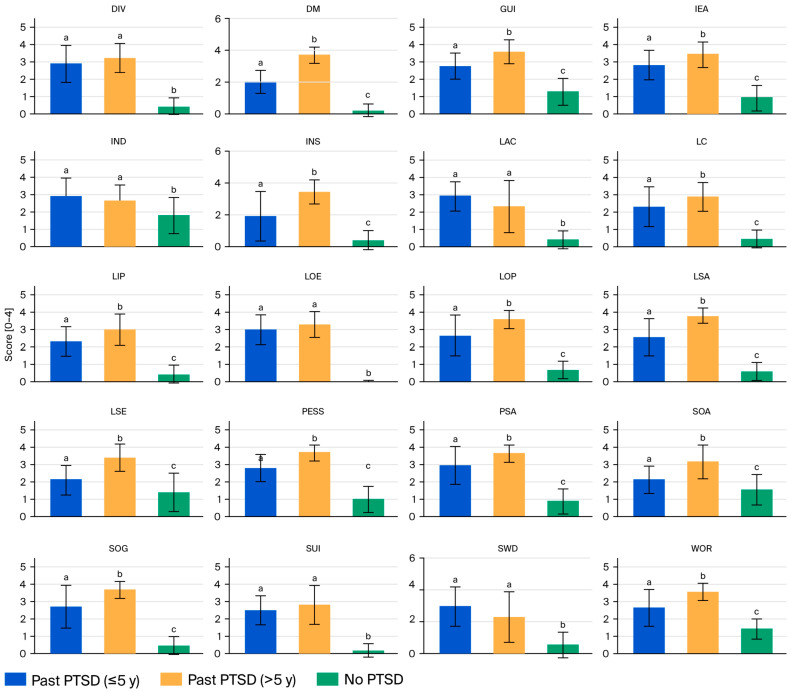
Mean scores with standard deviation error bars for Cornell Dysthymia Rating Scale (CDRS) domains across PTSD status groups with post hoc differences denoted using compact letter display. The domains are defined as follows: DM = depressed mood; LIP = lack of interest or pleasure; PESS = pessimism; SUI = suicidal ideation; LSE = low self-esteem; GUI = guilt; LC = lack of control; SWD = social withdrawal; IND = indecisiveness; LAC = low attention/concentration; PSA = psychic anxiety; SOA = somatic anxiety; WOR = worry; IEA = irritability or excessive anger; SOG = somatic general; LOP = low productivity; LOE = low energy; LSA = low sexual interest/activity; INS = insomnia; DIV = diurnal variation. For characteristics with significant differences (*p* < 0.05), groups sharing the same superscript letter (a, b, c) do not differ significantly, while different letters indicate significant differences based on post hoc testing.

**Table 1 ijms-26-10214-t001:** Demographic and clinical characteristics of participants by PTSD status.

Characteristic	Total (N = 92)	Past PTSD (≤5 y) (N = 33)	Past PTSD (>5 y) (N = 31)	No PTSD (Control) (N = 28)	*p*-Value
Demographic characteristics					
Age, years, median (IQR)	34.0 (28.0–41.0)	34.0 (31.0–41.0)	36.0 (29.5–41.0)	33.5 (24.3–41.5)	0.524
Employment under hazardous conditions, years, median (IQR)	10.0 (6.0–14.0)	11.0 (7.0–14.0)	10.0 (7.5–15.0)	10.0 (3.0–14.0)	0.418
Psychotherapy characteristics					
Psychotherapy, n (%)	59 (63.4)	25 ^a^ (75.8)	25 ^a^ (80.6)	9 ^b^ (32.1)	<0.001
Duration of psychotherapy, months, median (IQR)	2.0 (1.0–3.0)	3.0 ^a^ (2.0–4.0)	2.0 ^a^ (1.0–3.0)	1.0 ^b^ (0.0–2.0)	0.012
Type of psychotherapy, n (%)	59				0.353
CBT	47 (79.7)	22 (88.0)	19 (76.0)	6 (66.7)	
Psychodynamic	8 (13.6)	3 (12.0)	3 (12.0)	2 (22.2)	
Other	4 (6.8)	0 (0.0)	3 (12.0)	1 (11.1)	
Self-Destructive behaviors					
Self-destructive behaviors, n (%)	50 (54.9)	13 ^a^ (39.4)	26 ^b^ (86.7)	11 ^a^ (39.3)	<0.001
Type of self-destructive behavior, n (%)					0.683
Self-harm	30 (60.0)	9 (69.2)	14 (53.8)	7 (63.6)	0.552
Tattoos	9 (18.0)	1 (7.7)	5 (19.2)	3 (27.3)	0.552
Restrictive eating habits	3 (6.0)	0 (0.0)	3 (11.5)	0 (0.0)	0.180
Other	8 (16.0)	3 (23.1)	4 (15.4)	1 (9.1)	0.552

Notes: Data are presented as median (IQR) values for continuous variables and n (%) for categorical variables. *p*-values were calculated using the Kruskal–Wallis rank sum test for continuous variables and Pearson’s Chi-squared test or Fisher’s exact test for categorical variables. Individual *p*-values for types of psychotherapy and self-destructive behaviors were computed using Fisher’s exact test. Post hoc pairwise comparisons were performed using the Dunn test with Holm-Bonferroni correction and Chi-squared test; groups with different superscript letters (a, b) differ significantly (*p* < 0.05), while those sharing the same letter do not. Total N = 92 (sum of participants across groups: 33 + 31 + 28). Some characteristics have smaller sample sizes due to missing data (e.g., psychotherapy: N = 59; self-destructive behaviors: N = 50). IQR = interquartile range; CBT = cognitive behavioral therapy; PTSD = post-traumatic stress disorder.

**Table 2 ijms-26-10214-t002:** Biomarker levels across PTSD status groups.

Biomarker	Total (N = 92)	Past PTSD (≤5 y) (N = 33)	Past PTSD (>5 y) (N = 31)	No PTSD (Control) (N = 28)	*p*-Value	Post Hoc
SUMO1, ng/mL	11.40 (4.47–34.63)	4.47(3.51–5.44) ^a^	11.40 (6.67–19.59) ^b^	34.63 (26.26–41.76) ^c^	<0.001	a < b < c (all *p* < 0.001)
MDA, nmol/mL	8.37 (3.13–23.50)	23.50 (18.66–30.71) ^a^	8.37 (5.57–12.72) ^b^	3.13 (2.28–3.65) ^c^	<0.001	a > b, a > c (*p* < 0.001), b > c (*p* = 0.001)
CX3CL1, ng/mL	8.60 (3.59–30.13)	30.13 (22.66–38.27) ^a^	8.60 (6.02–15.69) ^b^	3.59 (3.09–4.96) ^c^	<0.001	a > b > c (all *p* < 0.001)
UCHL1, ng/mL	11.18 (3.98–30.54)	30.54 (19.92–36.16) ^a^	11.18 (6.38–16.69) ^b^	3.98 (3.04–4.67) ^c^	<0.001	a > b > c (all *p* < 0.001)

Notes: Data are presented as median (IQR) values. *p*-values were calculated using the Kruskal–Wallis rank sum test, with *p* < 0.001 indicating significant differences across groups for all biomarkers. Post hoc pairwise comparisons were performed using the Dunn test with Holm-Bonferroni correction; groups with different superscript letters (a, b, c) differ significantly (*p* < 0.05), while those sharing the same letter do not. The lowercase letters in the Post hoc column indicate the direction of the difference (e.g., a > b > c indicates Past PTSD ≤ 5 y > Past PTSD > 5 y > No PTSD). IQR = interquartile range; PTSD = post-traumatic stress disorder.

**Table 3 ijms-26-10214-t003:** Cornell Dysthymia Rating Scale (CDRS) scores across PTSD status groups.

Domain	Total (N = 92)	Past PTSD (≤5 y) (N = 33)	Past PTSD (>5 y) (N = 31)	No PTSD (Control) (N = 28)	*p*-Value	Post Hoc
DM	2.02 (1.47)	2.00 (0.71) ^a^	3.68 (0.48) ^b^	0.18 (0.39) ^c^	<0.001	a < b, c < a (all *p* < 0.001)
LIP	1.87 (1.37)	2.30 (0.85) ^a^	2.97 (0.87) ^b^	0.39 (0.50) ^c^	<0.001	a < b (*p* = 0.008), c < a (*p* < 0.001), c < b (*p* < 0.001)
PESS	2.53 (1.37)	2.82 (0.77) ^a^	3.71 (0.46) ^b^	1.00 (0.77) ^c^	<0.001	c < a < b (all *p* < 0.001)
SUI	1.84 (1.37)	2.48 (0.83) ^a^	2.81 (1.11) ^a^	0.18 (0.39) ^b^	<0.001	a > c, b > c (all *p* < 0.001)
LSE	2.30 (1.27)	2.12 (0.82) ^a^	3.39 (0.76) ^b^	1.39 (1.10) ^c^	<0.001	a > c, b > c (all *p* < 0.001), b > a (*p* = 0.020)
GUI	2.58 (1.24)	2.79 (0.74) ^a^	3.61 (0.67) ^b^	1.29 (0.76) ^c^	<0.001	c < a < b (all *p* < 0.001)
LC	1.88 (1.29)	2.30 (1.16) ^a^	2.87 (0.81) ^b^	0.43 (0.50) ^c^	<0.001	a > c, b > c (all *p* < 0.001)
SWD	1.93 (1.48)	2.94 (1.22) ^a^	2.29 (1.55) ^a^	0.54 (0.79) ^b^	<0.001	a > c, b > c (all *p* < 0.001)
IND	2.46 (1.09)	2.88 (1.08) ^a^	2.65 (0.88) ^a^	1.82 (1.02) ^b^	<0.001	a > c, b > c (all *p* < 0.001)
LAC	1.93 (1.45)	2.94 (0.83) ^a^	2.35 (1.50) ^a^	0.46 (0.51) ^b^	<0.001	a > c, b > c (all *p* < 0.001)
PSA	2.48 (1.37)	2.94 (1.09) ^a^	3.61 (0.50) ^b^	0.86 (0.71) ^c^	<0.001	a > c, b > c (all *p* < 0.001), b > a (*p* = 0.007)
SOA	2.28 (1.09)	2.12 (0.78) ^a^	3.16 (0.97) ^b^	1.54 (0.88) ^c^	<0.001	b > a, c (all *p* < 0.001), a > c (*p* = 0.020)
WOR	2.55 (1.22)	2.64 (1.03) ^a^	3.55 (0.51) ^b^	1.43 (0.57) ^c^	<0.001	c < a < b (all *p* < 0.001)
IEA	2.41 (1.32)	2.82 (0.81) ^a^	3.45 (0.72) ^b^	0.93 (0.72) ^c^	<0.001	a > c, b > c (*p* < 0.001), b > a (*p* = 0.004)
SOG	2.30 (1.47)	2.70 (1.24) ^a^	3.68 (0.48) ^b^	0.46 (0.51) ^c^	<0.001	c < a < b (all *p* < 0.001)
LOP	2.32 (1.38)	2.64 (1.17) ^a^	3.58 (0.50) ^b^	0.68 (0.48) ^c^	<0.001	c < a < b (all *p* < 0.001)
LOE	2.09 (1.47)	2.97 (0.85) ^a^	3.26 (0.73) ^a^	0.00 (0.00) ^b^	<0.001	a > c, b > c (all *p* < 0.001)
LSA	2.31 (1.47)	2.55 (1.06) ^a^	3.77 (0.43) ^b^	0.57 (0.50) ^c^	<0.001	c < a < b (all *p* < 0.001)
INS	1.92 (1.58)	1.88 (1.58) ^a^	3.42 (0.76) ^b^	0.43 (0.57) ^c^	<0.001	c < a < b (all *p* < 0.001)
DIV	2.20 (1.44)	2.88 (1.08) ^a^	3.23 (0.84) ^a^	0.43 (0.50) ^b^	<0.001	a > c, b > c (all *p* < 0.001)
CDRS Total	44.24 (21.81)	51.70 (7.38) ^a^	65.03 (6.89) ^b^	15.00 (3.57) ^c^	<0.001	c < a < b (all *p* < 0.001)

Notes: Data are presented as mean (SD) values. *p*-values were calculated using one-way ANOVA, with *p* < 0.001 indicating significant differences across groups for all domains. Post hoc pairwise comparisons were performed using Tukey’s HSD test; groups with different superscript letters (a, b, c) differ significantly (*p* < 0.05), while those sharing the same letter do not. The Post hoc column summarizes the direction of significant differences (e.g., a < b, c < a indicates Past PTSD ≤ 5 y < Past PTSD > 5 y, and No PTSD < Past PTSD ≤ 5 y). Each CDRS item is scored on a scale from 0 to 4 (0 = none, 1 = slight, 2 = mild, 3 = moderate, 4 = severe), with a total score range of 0 to 80, where higher scores indicate greater symptom severity. The domains are defined as follows: DM = depressed mood; LIP = lack of interest or pleasure; PESS = pessimism; SUI = suicidal ideation; LSE = low self-esteem; GUI = guilt; LC = lack of control; SWD = social withdrawal; IND = indecisiveness; LAC = low attention/concentration; PSA = psychic anxiety; SOA = somatic anxiety; WOR = worry; IEA = irritability or excessive anger; SOG = somatic general; LOP = low productivity; LOE = low energy; LSA = low sexual interest/activity; INS = insomnia; DIV = diurnal variation. CDRS = Cornell Dysthymia Rating Scale; PTSD = post-traumatic stress disorder; SD = standard deviation.

**Table 4 ijms-26-10214-t004:** Spearman correlation coefficients (ρ) between biomarkers and CDRS domains across PTSD status groups.

CDRS Domain	Past PTSD (≤5 y) (N = 33)				Past PTSD (>5 y) (N = 31)				No PTSD (Control) (N = 28)			
	SUMO1, ng/mL	MDA, nmol/mL	CX3CL1, ng/mL	UCHL1, ng/mL	SUMO1, ng/mL	MDA, nmol/mL	CX3CL1, ng/mL	UCHL1, ng/mL	SUMO1, ng/mL	MDA, nmol/mL	CX3CL1, ng/mL	UCHL1, ng/mL
DM	0.09	0.09	0.12	0.00	−0.28	−0.22	−0.28	−0.12	−0.18	0.00	0.08	0.21
LIP	−0.15	−0.33	−0.25	−0.36 *	−0.41 *	−0.33	−0.40 *	−0.25	−0.39 *	0.24	0.02	0.15
PESS	0.10	0.01	−0.18	−0.22	−0.26	−0.21	−0.26	−0.14	−0.39 *	−0.07	−0.27	−0.13
SUI	−0.10	−0.10	−0.24	−0.18	−0.17	−0.20	−0.18	−0.03	0.10	−0.03	0.27	0.10
LSE	−0.19	0.03	0.20	0.12	−0.30	−0.33	−0.35	−0.28	0.08	−0.01	0.13	0.11
GUI	0.09	0.22	0.25	0.30	−0.30	−0.25	−0.32	−0.19	−0.26	0.09	−0.21	−0.22
LC	−0.14	0.09	0.24	0.05	0.13	0.03	−0.06	0.04	−0.31	−0.13	−0.25	−0.38 *
SWD	−0.19	0.06	0.26	−0.01	0.01	−0.06	−0.02	0.13	0.17	0.29	0.13	0.11
IND	−0.33	−0.26	−0.04	−0.29	0.00	0.01	0.07	0.12	−0.04	−0.04	0.17	0.08
LAC	0.09	0.01	0.15	−0.01	−0.01	−0.03	−0.02	0.13	0.18	0.21	0.08	0.39 *
PSA	−0.30	−0.13	−0.09	−0.31	0.05	0.04	0.03	−0.04	0.18	0.01	0.00	−0.01
SOA	−0.14	0.02	0.02	−0.20	0.13	0.14	−0.22	0.07	−0.18	0.06	−0.08	0.05
WOR	−0.38 *	−0.15	−0.15	−0.33	0.08	−0.07	−0.01	0.13	0.14	0.05	0.22	0.04
IEA	−0.16	0.10	0.14	0.05	0.37 *	0.31	0.37 *	0.25	0.06	0.07	−0.32	−0.19
SOG	0.24	0.05	−0.01	0.18	0.14	0.18	0.10	0.33	−0.14	0.07	−0.19	−0.15
LOP	−0.29	−0.18	−0.17	−0.17	0.03	0.05	0.06	0.18	0.00	0.00	−0.26	−0.03
LOE	0.13	0.32	0.28	0.15	0.03	0.14	0.17	0.04	Not applied (Lack of variance)	Not applied	Not applied	Not applied
LSA	0.23	0.00	0.04	−0.17	−0.29	−0.24	−0.24	−0.19	−0.35	−0.24	−0.13	0.06
INS	−0.38 *	−0.05	0.17	−0.17	0.00	−0.06	0.00	0.01	−0.05	0.28	−0.13	0.08
DIV	0.31	0.26	0.20	0.45 **	0.34	0.33	0.41 *	0.27	0.10	0.06	0.08	0.17

Notes: Significance levels (unadjusted *p*-values) are indicated as follows: * 0.01 ≤ *p* < 0.05, ** 0.001 ≤ *p* < 0.01. Due to multiple comparisons (80 correlations per group), a Bonferroni-adjusted significance threshold of *p* < 0.000625 (0.05/80) applies; no correlations remain significant after adjustment.

## Data Availability

All data and analyses are available within the manuscript or upon request to the corresponding author.

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
