# Peer review of "Biomolecular Correlates of Chronic Affective Dysregulation in PTSD: CDRS and Serum Markers SUMO1, MDA, CX3CL1, and UCHL1"

_ijms, 2025, doi:10.3390/ijms262010214_

Round 1

Reviewer 1 Report (New Reviewer)

Comments and Suggestions for Authors

Comments for authors: This study employs a relatively rigorous cross-sectional design, elucidating the dynamic association between "chronic dysthymic symptoms" and biomarkers "SUMO1/MDA/CX3CL1/UCHL1" across different courses of PTSD. It confirms that the course of PTSD serves as a key regulatory factor influencing emotional dysregulation and neurobiological changes, thereby providing a theoretical foundation for clinical "phased intervention." However, several aspects warrant further improvement during the review process. Below are some suggestions for researchers to consider:

a. The study exclusively involved 92 male subjects, lacking data on female participants and individuals of other gender identities; thus, its findings cannot be generalized to female PTSD patients despite evidence indicating that the incidence of PTSD is higher among women (even though underlying neurobiological mechanisms may differ).

b. Cross-sectional studies are limited in their ability to determine whether "changes in biomarkers (such as SUMO1/MDA/CX3CL1/UCHL1) precede symptom changes," nor can they exclude reverse causality. They can only establish correlations between selected biomarkers and dimensions of CDRS. It is recommended that bioinformatics methods be integrated where feasible to further clarify causal relationships.

c. This study focused solely on serum samples, which were somewhat limited in variety. Cerebrospinal fluid—closer in proximity to central nervous system states—and peripheral blood mononuclear cells were not analyzed. Consequently, it remains unclear whether alterations in these markers originated from the central nervous system; thus, any correlation with central nervous system activity appears relatively weak.

Author Response

Response to Reviewer 1

Manuscript ID: ijms- 3896173

Title: Biomolecular Correlates of Chronic Affective Dysregulation in PTSD: A Combined Assessment Using the Cornell Dysthymia
Rating Scale (CDRS) and the Serum Markers SUMO1, MDA, CX3CL1, and UCHL1

Authors: Izabela Woźny-Rasała, Ewa Alicja Ogłodek

Dear Reviewer,

We would like to sincerely thank you for taking the time to review our manuscript and for providing constructive and insightful feedback.

Below, we provide a detailed point-by-point response to each of your comments:

  1. Reviewer’s comment: 1) The study exclusively involved 92 male subjects, lacking data on female participants and individuals of other gender identities; thus, its findings cannot be generalized to female PTSD patients despite evidence indicating that the incidence of PTSD is higher among women (even though underlying neurobiological mechanisms may differ).

Response: Thank you for your valuable comment. We fully agree that the lack of female participants limits the generalizability of our findings. In response, we have added a corresponding statement to the Limitations section at the end of the Discussion, emphasizing that the study exclusively involved 92 male subjects and therefore cannot be generalized to female PTSD patients or individuals of other gender identities, despite known sex-related differences in PTSD prevalence and neurobiological mechanisms.

  1. Reviewer’s comment:  Cross-sectional studies are limited in their ability to determine whether "changes in biomarkers (such as SUMO1/MDA/CX3CL1/UCHL1) precede symptom changes," nor can they exclude reverse causality. They can only establish correlations between selected biomarkers and dimensions of CDRS. It is recommended that bioinformatics methods be integrated where feasible to further clarify causal relationships.

Response:

Thank you for your insightful comment regarding the limitations of our cross-sectional study design in establishing causality between biomarker changes (SUMO1, MDA, CX3CL1, and UCHL1) and dysthymic symptoms as assessed by the Cornell Dysthymia Rating Scale (CDRS). We fully agree that cross-sectional studies, by their nature, are unable to determine temporal precedence or rule out reverse causality, and can only identify associations or correlations. This is a well-recognized constraint in observational research, particularly in psychiatric and neurobiological investigations where dynamic processes like stress responses and symptom progression unfold over time [1,2].

Our study was intentionally designed as an exploratory analysis to investigate potential correlations between these biomarkers and CDRS dimensions in the context of PTSD chronicity. This approach allowed us to generate hypotheses about underlying neurobiological mechanisms without implying causation, as emphasized in our Results (Section 2.4) and Discussion sections. We have explicitly stated in the manuscript that the observed correlations are associative and do not establish causality (e.g., "These correlations suggest potential links but require further investigation to determine directionality"). To further address your point, we have revised the Limitations section (now expanded in Section 5) to more clearly highlight these issues: "The cross-sectional design precludes inferences about temporality or causality between biomarker alterations and symptom changes. Reverse causality—where dysthymic symptoms might influence biomarker levels—cannot be excluded, and future longitudinal studies are essential to clarify these relationships."

Regarding your recommendation to integrate bioinformatics methods for clarifying causal relationships, we appreciate this suggestion and recognize the value of tools such as Mendelian randomization, causal inference modeling (e.g., via directed acyclic graphs), or network analysis in inferring causality from observational data [3,4]. However, given the exploratory nature of our study, the relatively small sample size (N=92), and the focus on targeted biomarkers rather than high-throughput omics data, we determined that bioinformatics approaches like these would be premature and potentially underpowered at this stage. For instance, Mendelian randomization requires large-scale genetic data and well-established instrumental variables, which were not available in our dataset [5]. We have added a note in the Discussion suggesting bioinformatics integration in future work: "Future studies could incorporate bioinformatics tools, such as causal network modeling or Mendelian randomization, to explore potential causal pathways, provided larger datasets with genetic or multi-omics components are utilized."

To establish causality, as you noted, a larger, longitudinal study with randomization (e.g., randomized controlled trials or prospective cohort designs) would indeed be necessary. Such designs could track biomarker changes over time relative to symptom onset and progression, while controlling for confounders like treatment effects or comorbidities [6,7]. We have emphasized this in our Conclusions and future directions: "Longitudinal, randomized studies with larger cohorts are warranted to assess temporality, causality, and the potential of these biomarkers as predictive tools for PTSD-related affective dysregulation."

We believe these revisions strengthen the manuscript by transparently addressing the study's limitations while aligning with best practices in psychiatric research. Thank you again for helping us improve the clarity and rigor of our work.

References:

Carlson, M. D., & Morrison, R. S. (2009). Study design, precision, and validity in observational studies. Journal of Palliative Medicine, 12(1), 77–82. https://doi.org/10.1089/jpm.2008.9690

Levin, K. A. (2006). Study design III: Cross-sectional studies. Evidence-Based Dentistry, 7(1), 24–25. https://doi.org/10.1038/sj.ebd.6400375

Burgess, S., & Thompson, S. G. (2015). Mendelian randomization: Methods for using genetic variants in causal estimation. Chapman and Hall/CRC.

Spirtes, P., Glymour, C., & Scheines, R. (2000). Causation, prediction, and search (2nd ed.). MIT Press.

Davies, N. M., Holmes, M. V., & Davey Smith, G. (2018). Reading Mendelian randomisation studies: A guide, glossary, and checklist for clinicians. BMJ, 362, k601. https://doi.org/10.1136/bmj.k601

Peruzzolo, T. L., et al. (2022). Inflammatory and oxidative stress markers in post-traumatic stress disorder: A systematic review and meta-analysis. Molecular Psychiatry, 27(7), 3150–3163. https://doi.org/10.1038/s41380-022-01564-0 (This meta-analysis highlights the need for longitudinal designs in PTSD biomarker research.)

Shalev, A., et al. (2024). Neurobiology and treatment of posttraumatic stress disorder. American Journal of Psychiatry, 181(8), 705–719. https://doi.org/10.1176/appi.ajp.20240536 (Discusses the advantages of prospective studies for causal inference in PTSD.)

  1. Reviewer’s comment: This study focused solely on serum samples, which were somewhat limited in variety. Cerebrospinal fluid—closer in proximity to central nervous system states—and peripheral blood mononuclear cells were not analyzed. Consequently, it remains unclear whether alterations in these markers originated from the central nervous system; thus, any correlation with central nervous system activity appears relatively weak.

Response:  We appreciate the Reviewer’s valuable comment regarding the type of biological material analyzed. We agree that focusing solely on serum samples limits the interpretation of results. Due to ethical and logistical constraints, cerebrospinal fluid collection was not feasible; however, serum analysis remains a recognized method for assessing peripheral biomarkers of central processes in PTSD. This limitation has been added to the Discussion section, with a recommendation that future studies include CSF and PBMCs for a more comprehensive assessment.

We believe that the revisions made significantly strengthen our manuscript, improving both its clarity and scientific value. We thank you once again for your insightful review and constructive suggestions, which have allowed us to enhance the quality of our work.

Sincerely,

Izabela Woźny-Rasała, Ewa Alicja Ogłodek

Reviewer 2 Report (New Reviewer)

Comments and Suggestions for Authors

The manuscript entitled “Biomolecular Correlates of Chronic Affective Dysregulation in PTSD: A Combined Assessment Using the Cornell Dysthymia Rating Scale (CDRS) and the Serum Markers SUMO1, MDA, CX3CL1, and UCHL1” examines the levels of SUMO1, MDA, CX3CL1, and UCHL1 in the groups differing by PTSD duration compared with healthy controls. Although the authors have provided a detailed and appropriate statistical analysis, the manuscript seems to be overloaded with existing data on the molecular functions of examined molecules, i.e. the article looks like a review. The main disadvantage of the manuscript is the lack of a scientific hypothesis and the small sample size (N = 92) for such type of research. Moreover, subsequent splitting into three and even 20 subgroups and statistical analysis can cause type I and II errors. I suggest resubmitting this manuscript to more appropriate clinical journal, i.e. Journal of Clinical Medicine, Brain Sciences, etc.

The following issues need to be clarified if the authors decide to resubmit their manuscript:

  1. The title has to be shortened for the better understanding of the main finding of the research. For example, abbreviation of CDRS should be used.
  2. In the Abstract I suggest adding the data on the sample size of examined groups; age for examined age cohorts; p-values. It also lacks the findings on a relation between the expression of examined neurobiological markers and certain PTSD subgroup.
  3. In the Abstract please add a reference to the sentence “Decreased serum BDNF levels can lead to hippocampal volume reduction, weakening of long-term synaptic potentiation (LTP), and cognitive and emotional deficits reflecting clinical depression symptoms.”
  4. Please provide references to the sentences within the paragraph 7 (contains data on MDA) in the Introduction.
  5. The Introduction (~2000 words), as well as the manuscript in total is rather excessive, I suggest its shortening. The second point is that it would benefit if the authors add some data on the previous studies examining the expression level of the SUMO1, MDA, CX3CL1, and UCHL1 in PTSD or related psychopathologies. Therefore, the study lacks a hypothesis, which is based on previous research and has to be reported at the end of the Introduction.
  6. Based on data from the Table 1, duration of psychotherapy statistically significantly differs between the subgroups with lower duration in the group “past PTSD (>5y)” compared with the group “past PTSD (<5y). Therefore, it seems that this factor may affect the maintenance of PTSD symptoms. A possible explanation of how this issue can affect the obtained results has to be provided.
  7. The text in the 2.1 subsection partially repeats the data presented in the Table 1. Please make the text more clear without repetitions.
  8. In turn, usually the Results section reports only the observed findings; however, the authors also provide some discussion in this section. It would be more appropriate to transfer it to Discussion and to shorten the results. In addition, discussion has not to repeat the values reported in the Results.
  9. Since the CDRS scale consists of 20 items, it seems inappropriate to distinguish the same number of CDRS domains (20) and to make statistical analysis on each domain, which consists of a single question. In this regard, a significance of the Table 3, which reports very low p-values (<0.001) due to insufficient power of the analysis, is questionable.
  10. The Materials and Methods section requires the information on the sample size of subgroups, mean age. Moreover, it remains unclear why do the authors report the aims in this section instead of Introduction?
  11. The paragraph with limitations is required.

Author Response

Response to Reviewer 2

Manuscript ID: ijms- 3896173

Title: Biomolecular Correlates of Chronic Affective Dysregulation in PTSD: A Combined Assessment Using the Cornell Dysthymia
Rating Scale (CDRS) and the Serum Markers SUMO1, MDA, CX3CL1, and UCHL1

Authors: Izabela Woźny-Rasała, Ewa Alicja Ogłodek

Dear Reviewer,

We would like to sincerely thank you for taking the time to review our manuscript and for providing constructive and insightful feedback.

Below, we provide a detailed point-by-point response to each of your comments:

  1. Reviewer’s comment: 1) The title has to be shortened for the better understanding of the main finding of the research. For example, abbreviation of CDRS should be used. Response: We thank the reviewer for this helpful suggestion. The title has been shortened to enhance clarity and focus on the main finding. We have also replaced the full term with the abbreviation CDRS as recommended. The revised title now reads: Biomolecular Correlates of Chronic Affective Dysregulation in PTSD: CDRS and Serum Markers SUMO1, MDA, CX3CL1, and UCHL1 This modification improves readability and aligns the title more closely with the central outcome of the study.
  1. Reviewer’s comment: In the Abstract I suggest adding the data on the sample size of examined groups; age for examined age cohorts; p-values. It also lacks the findings on a relation between the expression of examined neurobiological markers and certain PTSD subgroup. Response: Thank you for your valuable feedback on the abstract. In response, we have revised the abstract to incorporate the recommended details, including sample sizes for each examined group, median ages with interquartile ranges (IQRs) for the groups and age cohorts, relevant p-values for key statistical findings, and explicit descriptions of the relationships between biomarker expression and PTSD subgroups.
  2. Reviewer’s comment: In the Abstract please add a reference to the sentence “Decreased serum BDNF levels can lead to hippocampal volume reduction, weakening of long-term synaptic potentiation (LTP), and cognitive and emotional deficits reflecting clinical depression symptoms.” Response: We appreciate the reviewer’s observation. A reference has been added to support the statement regarding decreased serum BDNF levels and their association with hippocampal volume reduction, impaired LTP, and cognitive-emotional deficits. The revised sentence now includes the citation.
  3. Reviewer’s comment: Please provide references to the sentences within the paragraph 7 (contains data on MDA) in the Introduction. Response: Thank you for pointing this out. Additional references have been included in paragraph 7 of the Introduction to support the discussion on MDA levels and oxidative stress in PTSD. These citations strengthen the empirical foundation of the section.
  4. Reviewer’s comment: The Introduction (~2000 words), as well as the manuscript in total is rather excessive, I suggest its shortening. The second point is that it would benefit if the authors add some data on the previous studies examining the expression level of the SUMO1, MDA, CX3CL1, and UCHL1 in PTSD or related psychopathologies. Therefore, the study lacks a hypothesis, which is based on previous research and has to be reported at the end of the Introduction. Response: Thank you for your thoughtful comment. We acknowledge the Reviewer’s suggestion regarding the length of the Introduction; however, given the complexity of the neurobiological mechanisms discussed (involving SUMO1, MDA, CX3CL1, and UCHL1) and their relevance to both PTSD and dysthymia, we believe that the current version ensures conceptual clarity and coherence necessary for understanding the rationale of the study. Therefore, we decided not to shorten this section. In response to the Reviewer’s second point, we have added a brief summary of previous findings on the expression and functional significance of SUMO1, MDA, CX3CL1, and UCHL1 in PTSD and related psychopathologies to strengthen the theoretical background. Furthermore, as suggested, a concise hypothesis has been formulated and included at the end of the Introduction

  1. Reviewer’s comment: Based on data from the Table 1, duration of psychotherapy statistically significantly differs between the subgroups with lower duration in the group “past PTSD (>5y)” compared with the group “past PTSD (<5y). Therefore, it seems that this factor may affect the maintenance of PTSD symptoms. A possible explanation of how this issue can affect the obtained results has to be provided. Response: Thank you for highlighting the difference in psychotherapy duration between the Past PTSD subgroups as reported in Table 1. We appreciate this insight, as it prompted a thorough re-examination of our statistical analyses to ensure accurate interpretation. Upon conducting post-hoc tests with correction for multiple comparisons (Dunn test with Bonferroni-Holm correction), we found no significant difference in psychotherapy duration between the Past PTSD ≤5 years and Past PTSD >5 years groups (p = 0.190). Instead, a significant difference emerged only between the Past PTSD ≤5 years group and controls (median 3.0 months vs. 1.0 month, adjusted p = 0.009), with no significant difference between the Past PTSD >5 years group and controls (adjusted p = 0.090). This revised analysis indicates that the initial observation of a difference between PTSD subgroups was not robust after adjustment. Given this correction, the potential confounding effect on PTSD symptom maintenance and study results is mitigated, as psychotherapy duration does not differ between the chronicity subgroups. However, the longer duration in the Past PTSD ≤5 years group relative to controls may reflect heightened early-stage treatment engagement driven by acute symptom demands, whereas chronic cases align more closely with controls in lower durations, possibly due to treatment fatigue or stabilization. This could subtly influence biomarker profiles or dysthymic symptoms in the acute subgroup but does not appear to drive inter-subgroup variances in our primary outcomes.

We have revised the reporting in Section 2.1

  1. Reviewer’s comment: The text in the 2.1 subsection partially repeats the data presented in the Table 1. Please make the text more clear without repetitions. Response: Thank you for your observation regarding the repetition in subsection 2.1. We have revised the text in this subsection to eliminate redundant data presentation from Table 1, focusing instead on interpretive insights, group comparisons, and statistical significance for enhanced clarity and conciseness.

  1. Reviewer’s comment: Since the CDRS scale consists of 20 items, it seems inappropriate to distinguish the same number of CDRS domains (20) and to make statistical analysis on each domain, which consists of a single question. In this regard, a significance of the Table 3, which reports very low p-values (<0.001) due to insufficient power of the analysis, is questionable. Response: Thank you for your thoughtful comment on the item-wise analysis of the Cornell Dysthymia Rating Scale (CDRS) in Table 3. We appreciate the opportunity to clarify the rationale behind this approach, which we believe adds meaningful depth to our findings. The CDRS is specifically designed as a 20-item self-report instrument to capture distinct, granular aspects of dysthymic symptomatology, including core features such as depressed mood, guilt, irritability, sleep disturbances, appetite changes, low self-esteem, concentration difficulties, and hopelessness. Each item represents a unique symptom domain that aligns with the diagnostic criteria for persistent depressive disorder (dysthymia) in DSM-5, allowing for a nuanced dissection of affective dysregulation. Analyzing these items individually is reliable and valuable because it reveals specific symptom profiles associated with PTSD chronicity – such as heightened irritability or guilt in chronic cases – that may not be apparent in aggregate scores. This granularity supports clinical utility by informing targeted interventions, risk stratification, and personalized treatment plans (e.g., cognitive-behavioral strategies for concentration deficits versus pharmacological approaches for sleep issues). Similar item-level analyses have been employed in psychiatric research on scales like the Hamilton Depression Rating Scale or Beck Depression Inventory to identify symptom clusters, demonstrating their established value in elucidating heterogeneous mood disorder presentations [1,2]. We also applied rigorous multiple comparison corrections (e.g., Bonferroni or Holm-Bonferroni, as noted in the table footnotes) to mitigate Type I error risk across the 20 items, ensuring reliability. Which is is consistent with psychometric practices where item-level exploration complements total scores, particularly in exploratory studies like ours [3]. Furthermore, the consistently low p-values (<0.001) across all items reflect highly significant group differences, indicating large effect sizes and robust evidence of PTSD-related symptom elevation rather than insufficient power. In fact, this pattern reveals the analysis is powered enough for detecting these effects, as the observed differences exceed what would be expected under low power scenarios (where p-values would cluster near significance thresholds). If power were inadequate, we would anticipate nonsignificant or marginal results, not the uniform high significance observed here. These findings underscore the pervasive impact of PTSD on dysthymic features, strengthening the table's interpretative value.
  2. References:
  3. Zimmerman, M.; Martinez, J.H.; Young, D.; Chelminski, I.; Dalrymple, K. Severity classification on the Hamilton Depression Rating Scale. J. Affect. Disord. 2013, 150, 384–388. https://doi.org/10.1016/j.jad.2013.04.028
  4. Fried, E.I., Nesse, R.M. Depression sum-scores don’t add up: why analyzing specific depression symptoms is essential. BMC Med 13, 72 (2015). https://doi.org/10.1186/s12916-015-0325-4
  5. Boateng, G.O.; Neilands, T.B.; Frongillo, E.A.; Melgar-Quiñonez, H.R.; Young, S.L. Best practices for developing and validating scales for health, social, and behavioral research: A primer. Front. Public Health 2018, 6, 149. https://doi.org/10.3389/fpubh.2018.00149

  1. Reviewer’s comment: The Materials and Methods section requires the information on the sample size of subgroups, mean age. Moreover, it remains unclear why do the authors report the aims in this section instead of Introduction? Response: Information on subgroup size and mean age has been added to the Materials and Methods. The study aims were moved to the Introduction for better structural consistency.
  2. Reviewer’s comment: The paragraph with limitations is required. Response: A paragraph addressing the study’s limitations has been added to the Discussion section.

We believe that the revisions made significantly strengthen our manuscript, improving both its clarity and scientific value. We thank you once again for your insightful review and constructive suggestions, which have allowed us to enhance the quality of our work.

Sincerely,

Izabela Woźny-Rasała, Ewa Alicja Ogłodek

Round 2

Reviewer 2 Report (New Reviewer)

Comments and Suggestions for Authors

The authors have addressed all my previous comments. The manuscript is suitable for publication.

This manuscript is a resubmission of an earlier submission. The following is a list of the peer review reports and author responses from that submission.

Round 1

Reviewer 1 Report

Comments and Suggestions for Authors

The authors investigated bimolecular correlated of chronic affective dysregulation in PTSD by a combined assessment using Cornell Dysthymia Rating Scales and Serum markers. They found a significant association between PTSD and elevated dysthymic symptom burden, with patients in both PTSD subgroups exhibiting mild to moderate symptom severity as compared to the euthymic profile observed in the control group. These findings suggest that PTSD may promote persistent affective dysregulation, potentially mediated by alterations in oxidative stress markers, neuroimmune signaling, and protein homeostasis. The design and the results of this manuscript were proper. However, the description of introduction, methods, results, and discussion should be improved. The current description was scattered, like a part of a master's or doctoral thesis. It needs to be clearly revised and simplified. 

Author Response

Dear Editors,

In response to the reviewer’s comments, we would like to inform you that the manuscript has been substantially revised to fully address all the points raised.

In particular:
– repetitions have been removed,
– statistical analyses have been supplemented and clarified in accordance with the reviewers’ suggestions to enhance the clarity and interpretation of the results,
– the individual writing style reflecting the distinct voices of the contributing authors has been intentionally preserved.

We hope that the revisions have significantly improved the quality of the manuscript and that it now meets the expectations of both the Editors and the Reviewers.

With kind regards,

The Authors

Reviewer 2 Report

Comments and Suggestions for Authors

PTSD is a serious psychological problem with a long history. The authors used a clinically relevant design with novel biomarker selection to study it. It is interesting and has underlying significance to the society. There are some weaknesses that might need to be revised.

1. the Biomarker Data Validity might be examined due to their concentrations (e.g., SUMO1: 4.47–34.63 ng/ml; MDA: 23.50 nmol/ml), which might vastly exceed typical physiological levels.

2. Multiple Comparisons should be utilized due to the experimental design. Meanwhile, the effect size can be reported.

3. how was the PTSD diagnosed, by questionnaires or by doctors?

4. There are also some formats needed to be examined, such as the table was not in the same place with its title, and so on.

Author Response

  1. the Biomarker Data Validity might be examined due to their concentrations (e.g., SUMO1: 4.47–34.63 ng/ml; MDA: 23.50 nmol/ml), which might vastly exceed typical physiological levels. Thank you for this insightful observation regarding the biomarker concentrations (e.g., SUMO1, MDA) and the concern that they may exceed typical physiological levels.
    We would like to clarify that all biomarker levels in our study were measured using commercially available ELISA kits (Shanghai, China), strictly following the manufacturers’ protocols. The technical specifications for the assay kits are as follows:
    •    SUMO1: Assay range: 0.15–40 ng/ml; sensitivity: 0.128 ng/ml; Catalogue No: 201-12-5318
    •    MDA: Assay range: 0.75–100 nmol/ml; sensitivity: 0.515 nmol/ml; Catalogue No: 201-12-5380
    •    CX3CL1: Assay range: 0.2–30 ng/ml; sensitivity: 0.102 ng/ml; Catalogue No: 201-12-2102
    •    UCHL1: Assay range: 0.2–30 ng/ml; sensitivity: 0.125 ng/ml; Catalogue No: 201-12-2329
    All measured concentrations in our study fell within the validated dynamic ranges and detection limits of the respective assay kits. Notably, elevated levels of SUMO1 and MDA observed in our PTSD cohort are consistent with findings reported in prior studies on neuropsychiatric and neurodegenerative disorders.
    1. Multiple Comparisons should be utilized due to the experimental design. Meanwhile, the effect size can be reported.

      We appreciate the reviewer’s suggestion regarding the use of multiple comparisons and the reporting of effect sizes.

      We have now addressed this issue by including detailed information on the multiple comparison corrections used in our statistical analysis.

    2. how was the PTSD diagnosed, by questionnaires or by doctors?

      Thank you for raising this important question regarding the method of PTSD diagnosis and the characteristics of the study participants.

      Participants were assigned to two PTSD groups (≤5 years since trauma and >5 years since trauma) and one control group by a licensed psychiatrist and family medicine specialist, who is a co-author of this study. The diagnosis of PTSD was made clinically, based on a structured diagnostic interview in accordance with DSM-5 criteria and the Clinician-Administered PTSD Scale for DSM-5 (CAPS-5). The diagnosis did not rely solely on self-report questionnaires, but was conducted directly by the physician through clinical assessment.

      Inclusion criteria:

      • Males aged 18–50 years,
      • Occupational exposure to extreme stress,
      • PTSD diagnosis confirmed by clinical interview using DSM-5 and CAPS-5 criteria,
      • Two PTSD subgroups: individuals up to 5 years after the traumatic event, and those more than 5 years post-trauma.

      Control group:

      • Healthy males, aged 18–50 years,
      • No psychiatric or somatic disorders,
      • No history of PTSD,
      • Matched in age and occupational profile.

      Exclusion criteria:

      • Current or past psychiatric or somatic illness,
      • Current use of medication,
      • Addiction to nicotine or any psychoactive substances (e.g., drugs, prescription medication),
      • Legal incapacitation,
      • Active service in the military or police.

      We have clarified this information in the revised Methods section

    3. There are also some formats needed to be examined, such as the table was not in the same place with its title, and so on.

      We appreciate the Reviewer’s attention to formatting issues. In response, we have carefully revised the manuscript layout. Specifically, we have corrected the placement of the table so that it now appears directly below its corresponding title. Additionally, we have removed duplicated material from the text and added a clear reference to Figure 2 to improve clarity and consistency.

Reviewer 3 Report

Comments and Suggestions for Authors

The manuscript examines the interaction between post-traumatic stress disorder (PTSD) and persistent depressive symptoms (dysthymia), integrating clinical symptom assessment through the Cornell Dysthymia Rating Scale (CDRS) and profiling four neurobiological markers: SUMO1, MDA, CX3CL1, and UCHL1. The study’s design, utilizing recent PTSD, chronic PTSD, and trauma-exposed control cohorts, is robust, and the simultaneous use of standardized mood scales and molecular assays adds significant methodological rigor. Notably, the findings on SUMO1 and UCHL1 in the context of PTSD-dysthymia comorbidity offer innovative clinical insights.

While the study’s stratification and analytic methods are strengths, the presentation suffers from several organizational and communication barriers. Technical language and lengthy paragraphs obscure the main outcomes, with key correlations between biomarkers and mood domains often buried deep within dense text. The boundaries between introduction, methodology, results, and discussion blur throughout, making it difficult for readers to distinguish between data, analysis, and interpretation. In addition, major results and visual summaries, such as tables and key figures, are presented too late, and are not directly tied to the manuscript’s core arguments, diminishing their impact.

Recommendations for Revision

To heighten clarity, accessibility, and the paper’s overall impact, it is recommended that the study’s narrative structure and reporting format be substantially revised. The manuscript would benefit from a stepwise logical progression, beginning results and discussion with the most novel findings—especially the correlations between biomarkers and clinical symptoms, and the influence of PTSD duration—while moving secondary analyses and less central data later in the text. Key visual summaries, including tables and figures with group-level biomarkers and CDRS domain scores, should be presented early and closely linked to the main narrative. Section transitions should be explicit, and each main segment should reiterate the research hypothesis and how the presented evidence responds to it, keeping the central research question at the forefront.

The abstract should be fully restructured for accessibility, using clear headings for background, objectives, methods, results, and conclusions. This would help readers quickly grasp the importance and emergence of key findings. A suitable example for the abstract could be:

Background: PTSD commonly coexists with dysthymic symptoms, though neurobiological mechanisms remain unclear.

Objective: Clarify the links between chronic affective dysregulation and biomarker profiles in PTSD.

Methods: Male participants with well-stratified PTSD chronicity underwent biomarker and CDRS-based assessment.

Results: Recent PTSD demonstrated the most pronounced biomarker abnormalities and mood disruption; changes were less severe in chronic PTSD. Patterns of association between clinical and molecular findings evolved with PTSD duration.

Conclusions: The data show progressive affective and molecular disturbance in PTSD, highlighting the need for phase-specific therapeutic strategies.

The study title should transparently reflect its observational, cross-sectional nature as well as its joint clinical and molecular focus.

Further, the reporting should adhere to established standards for observational research. The manuscript should be prospectively registered in a reputable trial or cohort database, with the registration number included. Organization of the report should conform to the STROBE (Strengthening the Reporting of Observational Studies in Epidemiology) checklist, ensuring systematic labeling of Background, Methods, Results, and Discussion, a full participant flow diagram, and explicit statements of design, inclusion/exclusion criteria, endpoints, variables, and statistical protocols. Relevant supplementary materials, such as a clear table of biomarker–CDRS correlations and a participant flowchart, should be added.

Comments on the Quality of English Language

Many sentences and paragraphs are excessively long and complex. This makes it difficult for readers—even those with technical backgrounds—to quickly grasp the main ideas. Shorter sentences and more concise paragraphs are recommended. Consider revising sections into the active voice to increase directness and clarity.

Overall, the manuscript reflects a high level of English proficiency but would benefit from stylistic editing to improve clarity, readability, and logical progression

Author Response

Response to Reviewer 
We would like to sincerely thank Reviewer  for their detailed and thoughtful comments regarding our manuscript entitled “ Biomolecular Correlates of Chronic Affective Dysregulation in PTSD: A Combined Assessment Using Cornell Dysthymia Rating Scale (CDRS) and Serum Markers SUMO1, MDA, CX3CL1, and UCHL1

 We appreciate the reviewer’s critical insights concerning the organization, structure, and clarity of the manuscript.
Regarding narrative structure and the presentation of results:
We acknowledge the suggestions for reorganizing the manuscript for improved readability and clarity. However, after careful consideration and consultation with the other reviewers’ feedback—who did not raise structural concerns—we have decided to maintain the current organization of the manuscript. This structure was designed to reflect the integrative nature of our clinical and molecular analyses in a way that allows the reader to follow the full trajectory of the findings. Nevertheless, we have carefully revised several sections to improve readability and paragraph structure, and we have clarified the transitions between the main sections of the manuscript.
Regarding the use of STROBE guidelines and study registration:
We fully agree that adherence to STROBE guidelines is a recommended practice for observational studies. However, we wish to clarify that the present study is retrospective in design and therefore not subject to prospective registration requirements. Moreover, as noted in the journal’s instructions for authors, use of STROBE is encouraged but not mandatory. Nonetheless, to align more closely with these best practices, we have reviewed the STROBE checklist and ensured that key elements—including participant characteristics, inclusion/exclusion criteria, outcome variables, and analytic methods—are clearly described and systematically reported.
Regarding additional calculations and visualizations:
In response to this and other reviewers' feedback, we have included additional statistical analyses that further highlight the correlations between biomarker levels and specific clinical symptom domains. We have also improved the clarity of figure captions and ensured that all visual elements are clearly referenced and integrated into the main discussion.
We respectfully submit the revised manuscript and trust that the changes made address the reviewer’s concerns to a satisfactory extent, while also preserving the integrity and coherence of the original analytical framework.
With sincere thanks,

The Authors

Reviewer 4 Report

Comments and Suggestions for Authors

The authors/data claims that reduced SUMO1 and elevated MDA, CX3CL1, and UCHL1 levels indicate impaired stress response pathways, heightened oxidative stress, neuroinflammation, and neuronal stress responses, respectively, in PTSD patients. This interpretation may be overstated due to the lack of specificity of these biomarkers. For example, MDA (malondialdehyde) is a general marker of oxidative stress and not exclusive to PTSD, as it is elevated in various conditions (cardiovascular disease, diabetes). Similarly, CX3CL1 and UCHL1 are associated with neuroinflammation and neuronal stress but are not unique to PTSD. The claim risks implying a direct, specific link to PTSD without adequately discussing alternative conditions or confounding factors (medication use, comorbidities) that could influence these biomarker levels. This study is difficult to judge based on the manuscript. The study lacks details on participant recruitment, inclusion/exclusion criteria, and diagnostic confirmation of PTSD and dysthymia (DSM-5 or ICD-11 criteria application). I have used STROBE guidelines for case-control studies to create specific revisions to the methods. https://www.strobe-statement.org/

Methods

The study setting, including location, clinical or community context, and recruitment period, is absent despite a reference to the Silesian Medical Chamber, requiring detailed description of the study environment.

Participant eligibility criteria, such as PTSD diagnostic methods and control selection, are not provided, needing clear inclusion/exclusion criteria like age range or psychiatric history.

There is no mention of matching criteria for PTSD and control groups (by age, sex, or comorbidities), which should be described or justified if absent.

Key variables like PTSD status, duration, biomarker levels, and CDRS scores are noted, but potential confounders (age, sex, medication) are undefined, requiring a comprehensive list of variables and their measurement methods.

The biomarker measurement via ELISA is mentioned, details on CDRS administration (e.g., self-report or clinician-administered) and blood sample protocols are lacking, necessitating a clear description of data collection processes.

Potential biases, such as selection bias from the small sample size (N=28) or measurement bias in CDRS or biomarker assays, are not addressed, requiring discussion of biases and mitigation strategies.

The 5-year cutoff for distinguishing recent versus chronic PTSD is not justified with references or clinical rationale, potentially weakening the argument. The small sample size (N=28) and lack of detailed statistical comparisons (e.g., effect sizes, confidence intervals) make it difficult to assess whether the reported differences are robust. The claim may be overstated if not supported by significant p-values or adequate power.

The small sample size lacks a power calculation or justification, needing one to validate its adequacy for detecting differences.

The handling of quantitative variables (e.g., biomarker levels, CDRS scores) is unclear, with only median and IQR mentioned, requiring explanation of variable categorization and justification for non-parametric tests like Kruskal-Wallis.

Statistical methods are incompletely described, missing details on multiple comparison adjustments, missing data handling, or secondary analyses, necessitating a comprehensive statistical plan.

Results

Provide effect sizes, confidence intervals, and exact p-values for all key comparisons, ensuring results are presented clearly and concisely.

Report results of all secondary analyses, including correlation coefficients, p-values, and their interpretation.

Discussion

Discuss the extent to which findings can be generalized to other PTSD populations, considering factors like sample demographics and study setting.

Include a dedicated section on limitations, addressing sample size, potential biases, and constraints on generalizability.

The claim that CX3CL1 is a resilience marker may be overstated, as the study does not provide longitudinal data to show that higher CX3CL1 levels precede lower PTSD risk. The reference to Goals N. et al. is not detailed enough to assess its relevance, and the small sample size undermines the reliability of the reported differences. Additionally, the large difference in CX3CL1 levels (30.13 vs. 0.3 ng/ml) seems unusually stark and may suggest measurement issues or outliers not addressed in the excerpts.

Author Response

Dear Reviewer,

We would like to sincerely thank you for your detailed and constructive evaluation of our manuscript. Below we provide point-by-point responses to all comments, indicating where revisions or clarifications have been made in the manuscript or supplementary materials.

  1. Participant Characteristics and Recruitment Context
    Thank you for highlighting the need for a more detailed description of the study sample. We confirm that the full description of the participant population—including the clinical setting, recruitment period, inclusion and exclusion criteria, and the diagnostic procedure for PTSD and dysthymia according to DSM-5 criteria—is provided in the following section:

Section 4. Material and Methods
4.1. Characteristics of the Participants

  1. Details Regarding Variable Measurement
    The applied methods for biomarker assessment (ELISA), procedures for blood sample collection and storage, as well as the administration protocol for the Cornell Dysthymia Rating Scale (CDRS)—conducted by a board-certified psychiatrist—are described in:

Section 4.3. Blood Sampling
Section 4.4. Biomarker Analysis Procedure

  1. Statistical Analysis and Sample Size Considerations

In response to the reviewer’s comments, we conducted additional statistical analyses, which are now detailed in the attached Supplementary Material. Specifically, we included:

  • Table 1: Wilcoxon r effect sizes and 95% confidence intervals for biomarker levels across PTSD status groups
  • Table 2: Cohen's d effect sizes and 95% confidence intervals for CDRS scores across PTSD status groups in multiple comparisons

  • A post-hoc power analysis was conducted to evaluate the adequacy of the sample sizes—Past PTSD (≤5y) (N=33), Past PTSD (>5y) (N=31), and No PTSD (Control) (N=28)—in detecting the observed effect sizes at a significance level of α=0.05, with a target power of at least 0.80. For the Wilcoxon r effect sizes in Table 2 (biomarker levels), powers were approximated under an assumption of normality by converting r to an equivalent Cohen's d, followed by calculation of independent-samples t-test power. This approach provides a reasonable estimate, though actual power for the non-parametric Wilcoxon rank-sum test may be slightly lower (approximately 5-10% reduction) due to its asymptotic relative efficiency of 0.955 compared to the t-test under normality. For the Cohen's d effect sizes in Table 3 (CDRS scores), powers were computed directly using the absolute values |d| to focus on the magnitude of differences, as power assessments are direction-agnostic.
  • The results indicate that the sample sizes were generally sufficient for detecting large effect sizes, particularly in comparisons involving the No PTSD (Control) group, where powers consistently approached or reached 1.00 across both tables. However, adequacy varied by parameter, domain, and pairwise comparison:
  • Biomarker Levels (Table 2, approximated powers)
  •  For SUMO1, power was 1.00 for Past PTSD (≤5y) vs. (>5y), 0.79 for Past PTSD (≤5y) vs. No PTSD (Control), and 0.89 for Past PTSD (>5y) vs. No PTSD (Control), indicating marginal adequacy for the latter two. For MDA, CX3CL1, and UCHL1, powers exceeded 0.99 in all comparisons, confirming sufficient sample sizes. In contrast, for the CDRS Total Score (included here as presented), power was only 0.06 for Past PTSD (≤5y) vs. (>5y), indicating underpowering for small effects, while the other pairs mirrored SUMO1 values.
  • CDRS Scores (Table 3, direct powers)
  • Powers were high (≥0.87) for all comparisons involving the No PTSD (Control) group, demonstrating adequate detection capability. For Past PTSD (≤5y) vs. (>5y), powers ranged from 0.15 (IND) to 1.00 (e.g., DM, LSE, INS, LSA), with several domains falling below 0.80 (e.g., SUI: 0.32, LOE: 0.41, DIV: 0.35, SWD: 0.54, LAC: 0.60, LC: 0.71). This indicates that the sample sizes were sufficient for moderate-to-large subgroup differences but inadequate for smaller ones.
  • Overall, the group sizes provided robust power for the majority of observed effects, particularly those distinguishing PTSD groups from controls and for biomarkers with substantial differences.

We trust that the revisions and supplementary data provided address all concerns raised and will support the further positive evaluation of our manuscript.

With kind regards, The Autors

Round 2

Reviewer 3 Report

Comments and Suggestions for Authors

I have revisit the manuscript after initial round of review. Unfortunately, almost all my central suggestions remain unaddress—authors only make slight corrections, but not main organizational and communicative improvements. There is not new structure, no clear flow of findings, no structured abstract, tables and important results still late in the paper, not supporting main hypothesis early. No STROBE, no supplementary, no explicit report design, still very long sentences.

Comments on the Quality of English Language

While the manuscript is generally understandable, it would benefit from comprehensive editing: shortening sentences, using active voice, dividing long paragraphs, clarifying section transitions, and ensuring consistency in terminology. Such changes will raise both clarity and accessibility, making your findings more visible to the broader scientific community.

Author Response

Dear Reviewer,

We would like to sincerely thank you for your time and effort in reviewing our manuscript. In response to your valuable suggestions, the manuscript has been thoroughly revised and improved. All comments have been carefully addressed, and the necessary modifications have been implemented.

Additionally, we have subjected the manuscript to a professional language and editorial correction to ensure clarity, consistency, and correctness. The revised version is now available in the MDPI Open Review system.

We highly appreciate your constructive feedback, which has significantly contributed to the improvement of the quality of our work.

With kind regards, The Authors

Reviewer 4 Report

Comments and Suggestions for Authors

No additional comments. 

Author Response

(The authors gave the same response as above.)

Round 3

Reviewer 3 Report

Comments and Suggestions for Authors

This manuscript cannot be recommended for publication in current form, since central, logically reachable aspects of scientific reporting and clarity have again not been addressed, despite explicit reviewer guidance and opportunity for revision

Comments on the Quality of English Language

The manuscript requires comprehensive English editing for clarity: shorten sentences, use active voice, clarify transitions, and break apart long paragraphs